# Human epidermal Langerhans cells induce tolerance and hamper T cell function upon tick-borne pathogen transmission

Johanna Strobl [1,2,9,10] ✉, Lisa Kleissl [1,2,9], Julia Eder [1,2], Sally Connolly [3], Tobias Frey [4], Laura Marie Gail [1,2], Aglaja Kopf [1,2], Sophie Weninger[1], Mateusz Markowicz[5], Pavlína Bartíková [6], Christian Freystätter[7], Klaus Schmetterer [4], Herbert Strobl [3], Hannes Stockinger [8], Michiel Wijnveld [8] & Georg Stary [1,2,10] ✉

Arthropods are ancient vectors of infectious disease that alter the immune environment of the skin during feeding. The epidermis and its immune sentinels, including Langerhans cells, are critical for protection against ectoparasitic arthropods such as ticks. Here, we investigate how human Langerhans cells respond to clinical and experimental tick bites and concomitant infection with the tick-borne bacterium Borrelia burgdorferi. Using imaging, migration assays, immune spheroid models, and single-cell transcriptomic analysis of patient samples, we show that tick bites and tick saliva reprogram Langerhans cells to increase migration into lymphatic tissues, adopt a tolerogenic state marked by specific transcriptional programs, reduced ability to induce pro-inflammatory helper T cells, and enhanced promotion of type 2 and regulatory T cell responses. This shift dampens protective immunity and helps explain how ticks and their pathogens evade host defense and achieve efficient transmission.

As a consequence of climate change and an expanded habitat of hard ticks, tick-borne diseases are emerging. *Ixodes ricinus*, the most common tick species in central Europe, is a transmission vector for a diverse range of pathogens including *Borrelia burgdorferi* species, which cause Lyme disease with characteristic erythema migrans, a highly prevalent skin infection[1,2]. Interestingly, tick salivary components transmitted to the human host during tick feeding possess immunosuppressive properties and may therefore increase transmission rates of tick-borne infections[3,4]. Indeed, we recently described rapidly occurring patterns of immunomodulation upon tick feeding on

human skin including a decrease in cutaneous dendritic cell (DC) numbers and modulation of cytokine production by lymphocytes[5].

Langerhans cells (LCs) are specialized skin-resident monocyte- and embryogenic precursor-derived antigen-presenting cells[6,7] and may provide both important pro-inflammatory and tolerogenic functions at the skin barrier[8,9]. Epidermal LCs belong to the first skin-resident immune cells that encounter tick-borne *B. burgdorferi* spirochetes[10]. It is well-established that active erythema migrans lesions contain decreased numbers of LCs, which is thought to be a consequence of *B. burgdorferi*-mediated LC activation and tissue

[1]Department of Dermatology, Medical University of Vienna, Vienna, Austria. [2]CeMM Research Center for Molecular Medicine of the Austrian Academy of Sciences, Vienna, Austria. [3]Division of Immunology, Otto Loewi Research Center, Medical University of Graz, Graz, Austria. [4]Department of Laboratory Medicine, Medical University of Vienna, Vienna, Austria. [5]Austrian Agency for Health and Food Safety (AGES), Vienna, Austria. [6]Biomedical Research Center – Institute of Virology, Slovak Academy of Sciences, Bratislava, Slovakia. [7]Department of Plastic, Reconstructive and Aesthetic Surgery, Medical University of Vienna, Vienna, Austria. [8]Institute for Hygiene and Applied Immunology, Center for Pathophysiology, Infectiology and Immunology, Medical University of Vienna, Vienna, Austria. [9]These authors contributed equally: Johanna Strobl, Lisa Kleissl. [10]These authors jointly supervised this work: Johanna Strobl, Georg Stary. ✉e-mail: johanna.strobl@meduniwien.ac.at; georg.stary@meduniwien.ac.at

emigration[11]. However, some studies imply interactions between LCs and tick saliva in (animal) model systems and indicate a decrease in the frequency of LCs around the sites of tick attachment independent of tick-borne pathogen transmission[5,12,13]. Investigations of in vitro-stimulated antigen presenting cells (APCs) activated with tick salivary antigens indicate that LCs play a role in mounting the immune response during tick infestation[14]. Furthermore, the adoptive transfer of *B. burgdorferi*-pulsed LCs and DCs into mice induced a protective immune response against *B. burgdorferi* and stimulated the production of specific antibodies in vivo[15]. The presence of LCs at the tick feeding site may thus be pivotal for the success of "tick vaccines", i.e., mRNA-based vaccines encoding tick salivary proteins to impair tick feeding[16] as well as Lyme disease vaccines that are currently under investigation[17]. Overall, the response of human epidermal LCs to tick bite and tick-borne pathogens remains ambiguous and warrants further investigation. In this study, we assess the role of human epidermal LCs in response to tick feeding and tick-borne pathogen transmission, with particular emphasis on how these cells are reprogrammed by tick saliva, how they migrate and interact with other immune cells, and how these processes shape subsequent adaptive immune activation and regulation, potentially influencing susceptibility to infection and the development of long-term protective immunity.

## Results

### Epidermal Langerhans cells decrease upon clinical and experimental tick bite

The epidermis is the tissue most densely populated by LCs and the first skin layer to encounter foreign antigens. To quantify the effect of tick feeding on epidermal LCs, we sampled skin punch biopsies of healthy donors presenting with tick bite without signs of tick-borne infection. We detected decreased numbers of CD1a⁺ (Fig. 1A, B) and CD207⁺ (Supplementary Fig. 1A, B) LCs in the epidermis after tick bites compared to healthy control skin sites (HC) of the same individuals. This effect was independent of the sampled body site of tick bites (Supplementary Fig. 1C), and we observed no increase in LCs presenting with ds-DNA breaks (gH2AX+ LCs, Supplementary Fig. 1D), suggesting that LCs are not killed upon tick feeding. To analyze the effects of tick feeding on human immune cells, we recently described an ex vivo human skin tick bite model[5]. Here, we use this model to inject *Ixodes ricinus* tick salivary gland extracts (SGE) into human skin explants (Fig. 1C). Consistent with our findings in clinical tick bite, we observed a decrease of epidermal LCs upon SGE injection (Fig. 1C, D, and Supplementary Fig. 1E). After injection with either SGE or PBS, similar numbers of LCs were found in the supernatant of the cultured full-thickness skin explants, indicating a lack of un-directed LC emigration from skin grafts in response to tick salivary molecules (Supplementary Fig. 1F). Epidermal sheets cultivated with SGE-supplemented media showed decreased LC networks (Supplementary Fig. 1G) and increased LC migration into the supernatant (Supplementary Fig. 1H) compared to epidermal sheets with media only, corroborating LC emigration from the epidermis after tick bite. To account for a potential replacement of LCs by other antigen-presenting cells, we next investigated HLA-DR-expressing cell numbers. Overall numbers and percentages of antigen-presenting cells did not change upon clinical or experimental tick bites (Supplementary Fig. 1I, J).

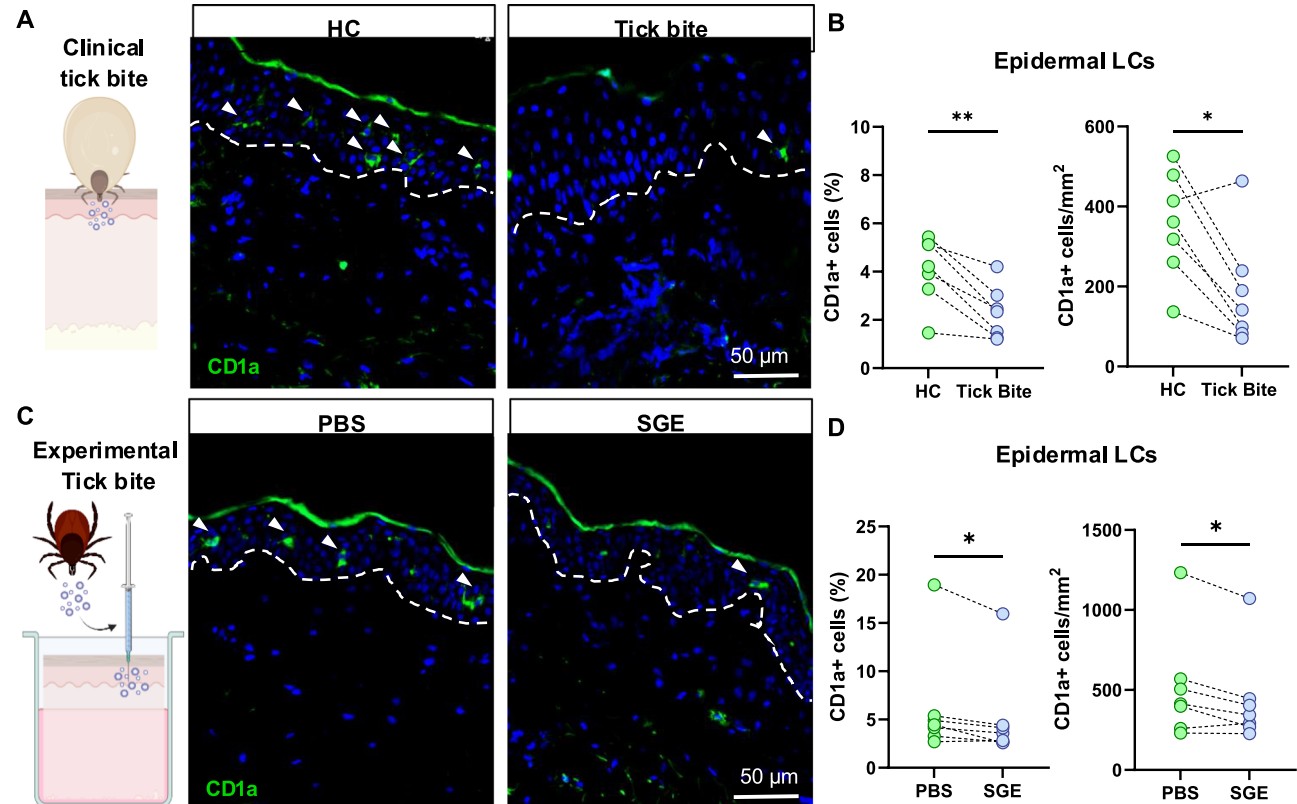

**Fig. 1 | Decreased numbers of epidermal Langerhans cells in skin upon clinical and experimental tick bite.** Graphical illustrations (left) and representative immunofluorescence images (right) of sections from clinical (**A**) and experimental (**C**) tick bite compared to patient-matched healthy control (HC) skin biopsies stained with CD1a (green) and DAPI (blue). Dashed lines separate dermis from epidermis, arrows indicate CD1a⁺ LCs in the epidermis. Quantification of relative and absolute CD1a⁺ epidermal LC numbers stained in clinical (*n* = 7) *P* = 0.0026, *P* = 0.0111 (**B**) and experimental (*n* = 7) *P* = 0.037, *P* = 0.0247 (**D**) tick bite skin biopsies. In **B**, **D**, 1 dot represents one patient, and dotted lines connect intraindividual samples. Illustrations (**A**, **C**) created in BioRender. Stary, G. (2025) https://BioRender.com/o1cvxpj. Statistical significance was calculated using paired Student's *t* test (two tailed). (*P* < 0.05; **P* < 0.01).

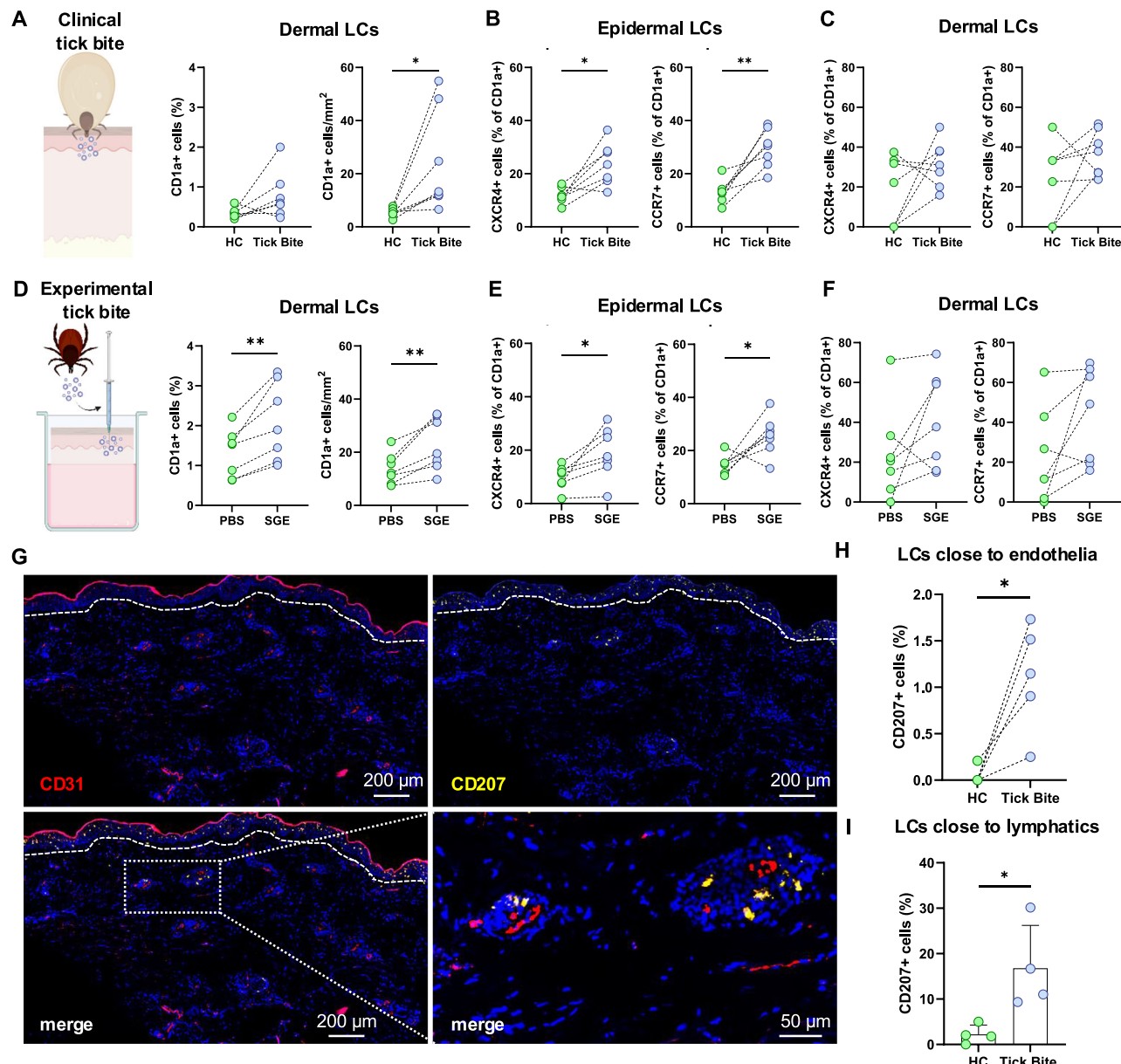

**Fig. 2 | Langerhans cells express homing markers and migrate to the dermis upon tick feeding.** Quantification of absolute and relative numbers of dermal CD1a⁺ LCs in clinical ($n = 7$) $P = 0.0323$, P = 0.037, $P = 0.0247$ (**A**) and experimental ($n = 7$) (**D**) tick bite skin biopsies. Epidermal (**B, E**) and dermal (**C, F**) CXCR4 and CCR7 protein expression in LCs after clinical ($n = 7$) $P = 0.0110$, $P = 0.0029$ (**B, C**) or experimental ($n = 7$) $P = 0.0054$, $P = 0.0072$ (**E, F**) tick bite. **G** IF images of CD207⁺ LCs (yellow) and CD31⁺ vessels (red). Quantification of LCs within or close to (distance <20 μm) CD31⁺ vessels ($n = 5$) $P = 0.0167$ (**H**) and podoplanin⁺ lymph vessel endothelial cells ($n = 4$, data are presented as mean values + SEM) $P = 0.0233$ (**I**). Dashed lines separate dermis from epidermis. **A–I** each dot represents one patient, and dotted lines connect intraindividual samples. *N* represent biological replicates. Statistical significance was calculated using paired (**A–H**) and unpaired (**I**) Student's *t* test (two tailed). (*$P < 0.05$; **$P < 0.01$). Graphic created in BioRender. Stary, G. (2025) https://BioRender.com/o1cvxpj.

## Langerhans cells up-regulate CXCR4 and CCR7 to migrate to dermis and lymph vessels after tick bite

We suspected a directed migration of LCs to lymphatic vessels in deeper skin structures and assessed spatial distribution of LCs in all skin layers of clinical tick bite-affected skin. We found significantly increased densities and absolute numbers of CD1a⁺ (Fig. 2A) and CD207⁺ LCs (Supplementary Fig. 2A) in the dermis compared to HC, indicating LC migration from epidermis into deeper skin layers.

Emigration of epidermal LCs occurs in two steps: during initial migration to the dermis, LCs express the chemokine receptor CXCR4 binding to CXCL12 on dermal fibroblasts[18,19]. Subsequently, LCs enter lymphatic vessels, which is dependent on the interaction of CCR7 with

its ligands CCL19 and CCL20[20]. Therefore, we mapped chemokine receptor expression on LCs in the skin and detected upregulation of CXCR4 and CCR7, particularly on epidermal LCs after tick bite (Fig. 2B, C, and Supplementary Fig. 2B–D). We corroborated these findings in our experimental tick bite model (Fig. 2D–F).

As increased CCR7 expression suggests LC migration to the lymphatics, we investigated spatial relations between LCs and CD31⁺ blood and lymph vessel endothelial cells in the dermis of tick bite-affected skin (Fig. 2G–I, and Supplementary Fig. 2E). The percentage of LCs in close proximity (distance ≤20 μm) to, or within CD31⁺ vessels, was significantly increased after tick bite (Fig. 2G, H). Similarly, increased numbers of LCs were observed inside or near (distance ≤20 μm)

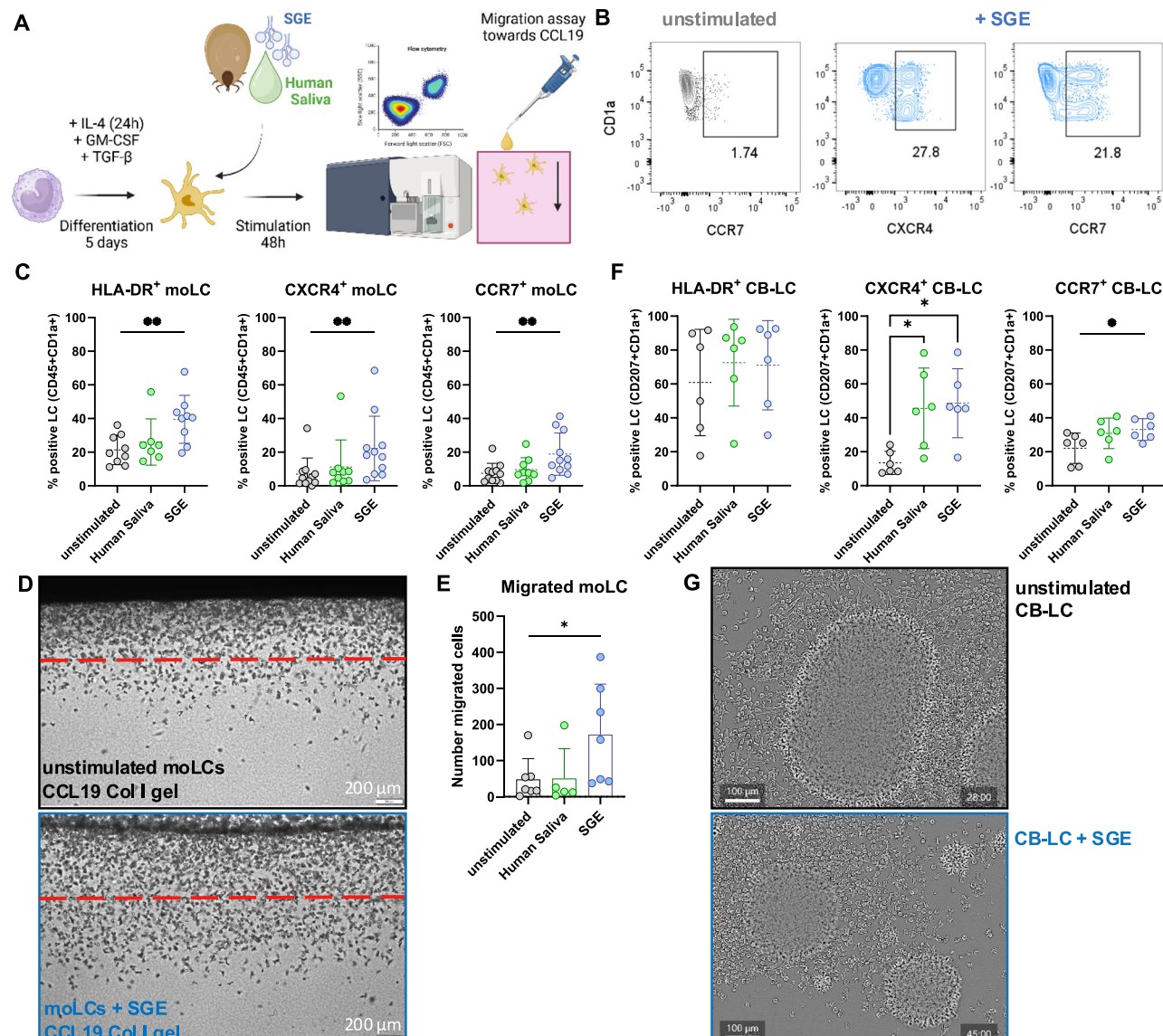

**Fig. 3 | Langerhans cells upregulate CXCR4 and CCR7 and emigrate to lymphatic vessels upon contact with tick salivary gland extracts. A** Conceptual overview of moLC differentiation and 48 h stimulation with SGE and human saliva. Created in BioRender. Stary, G. (2025) https://BioRender.com/ltj8ua7.
**B** Representative gating of CXCR4 and CCR7 expression on unstimulated CD1a⁺ moLCs (left panel) and after SGE stimulation (right). Expression levels of HLA-DR, CXCR4 and CCR7 on unstimulated (*n* = 11), human saliva (*n* = 9) and SGE (*n* = 11) stimulated moLCs *P* = 0.0012, *P* = 0.0040, *P* = 0.0076 (**C**) and CB-LCs (*n* = 6) *P* = 0.0481, *P* = 0.0252, *P* = 0.0167 (**F**). Representative images (**D**) of a collagen invasion assay of unstimulated or human saliva/SGE stimulated moLCs into a CCL19-supplemented collagen gel. **E** Quantification of migrated moLCs below threshold indicated by red dashed line in (**D**) after pulsing with SGE (*n* = 7), human saliva (*n* = 5) or left unstimulated (*n* = 7) *P* = 0.0224. **G** Representative images of live cell imaging performed on CB-LCs after simulation with SGE compared to unstimulated cells. Full videos can be found in supplementary material (unstimulated CB-LC in supplementary video 1, SGE-incubated CB-LC in supplementary video 2). **C**, **F**, **E** Data are plotted as mean values ± SD. *N* represent biological replicates. Statistical significance was calculated using Mixed-effects analysis (for moLCs) and RM one-way ANOVA (for CB-LCs) with Geisser-Greenhouse correction and Dunnett multiple comparison correction (*P < 0.05; **P < 0.01).

podoplanin⁺ lymphatic endothelial cells (Fig. 2I), with a tendency toward increased CCR7 expression on lymphatics-associated LCs (Supplementary Fig. 3A, B). Consistent with these findings, the mean distance between CD207⁺ and podoplanin⁺ cells was significantly decreased in tick bite biopsies (Supplementary Fig. 3C, D).

**Tick saliva contents induce Langerhans cell migration to the lymphatics**
To dissect whether LC emigration is mechanistically dependent on tick saliva, we exposed monocyte-derived LCs (moLCs) from healthy individuals to SGE or human saliva and analyzed invasion towards a CCL19-supplemented collagen gel (Fig. 3A). We found upregulation of CXCR4 and CCR7 on activated moLCs 48 h after stimulation with SGE, but not upon incubation with human saliva or in unstimulated controls (Fig. 3B, C). MoLC migration across a defined threshold in the collagen gel was sharply increased after SGE-stimulation (Fig. 3D, E). As moLCs phenotypically reflect LCs that are recruited to the skin after injury and inflammation[21], we also investigated the effect of tick saliva on the migration behavior of a steady-state LC phenotype using LCs derived from CD34⁺ cord blood progenitors (CB-LCs)[22]. Similar to moLCs, we observed a significant upregulation of CXCR4 and CCR7 on CB-LCs after stimulation with SGE, with a slightly higher propensity of CB-LCs to also react to human saliva (Fig. 3F). CB-LCs tend to form dense cluster networks when cultivated in vitro[22,23]. However, upon incubation with SGE, CB-LCs emigrated from large clusters to form several smaller clusters (Fig. 3G, and Supplementary Videos 1, and 2),

demonstrating an increased migration capacity. Transforming growth factor β (TGF-β) is a cytokine released by keratinocytes to maintain epidermal LC networks under homeostatic conditions[24]. We observed decreased TGF-β production by keratinocytes treated with SGE (Supplementary Fig. 3E), which may play a mechanistic role in LC emigration after tick feeding. Overall, our observations indicate that tick salivary contents inhibit TGF-β in keratinocytes, which is paralleled by activation and CXCR4 and CCR7-dependent emigration of LCs to dermal lymphatic vessels.

### Langerhans cells are activated and up-regulate tolerogenic transcription factors after tick salivary gland extract-stimulation

After emigration to lymphatic vessels, activated LCs exert their function by polarizing of T cells in draining lymph nodes. Depending on the activation of specific transcription factors, LCs may provide immunogenic or tolerogenic functions in response to antigen stimulation[25]. In response to pathogens, immunogenic transcription factors *IRF1* and *NFKB* lead to the production of pro-inflammatory cytokines including interferon-gamma (IFN-γ) and tumor necrosis factor-alpha (TNF-α). In a tolerogenic setting (e.g., healthy skin) *IRF4* and *IDO1* are active to facilitate tissue homeostasis[26]. We therefore investigated the expression of activation markers and transcription factors by moLCs and CB-LCs upon SGE stimulation (Fig. 4A). After exposure to foreign antigens, moLCs expressed higher levels of HLA-DR and CD11b after SGE stimulation, indicating their activation (Fig. 4B, and Supplementary Fig. 4C). However, SGE-stimulated, activated moLCs upregulated transcription factors typically expressed by tolerogenic LCs under homeostasis and not immunogenic transcription factors (Fig. 4C, E).

### Langerhans cells up-regulate tolerogenic transcription factors after challenge with tick-borne *B. burgdorferi*

As the function of LCs after tick bite is pivotal for an immunogenic response resulting in elimination of tick-borne pathogens, we next investigated whether addition of bacteria may abrogate the pro-tolerogenic polarization of LCs by SGE (Fig. 4A). Interestingly, co-incubation with the tick-borne pathogen *B. burgdorferi* induced a strong tolerogenic phenotype in moLCs, an effect that was further increased upon combined stimulation with SGE (Fig. 4D, and Supplementary Fig. 4A). Upon stimulation with *S. aureus*, which causes a pro-inflammatory immune response in human skin, we observed a strong increase in immunogenic IRF1+ and NFκB+ single- as well as double-positive moLC populations, that were absent in *B. burgdorferi*-stimulated cells (Fig. 4F, and Supplementary Fig. 4B). In addition, *S. aureus*-stimulated moLCs also expressed IDO1 (Supplementary Fig. 4A). Importantly, moLC treated with human saliva showed lower expression of the activation and migration markers HLA-DR, CCR7 and CXCR4 when compared to cells treated with SGE (Supplementary Fig. 4C). Expression of the transcription factors IDO1 and IRF4 was also increased in moLC treated with SGE but not with human saliva (Supplementary Fig. 4D). These polarization effects were less pronounced in CB-LCs, which are thought to be inherently tolerogenic in the skin[27,28]. While the expression of HLA-DR was significantly increased in CB-LC when treated with SGE (Supplementary Fig. 4E), we did not detect significant changes in the expression of tolerogenic transcription factors by CB-LCs upon stimulation with SGE, *S. aureus* or *B. burgdorferi* (Supplementary Fig. 4F, G). While stimulation with *S. aureus* increased the fraction of pro-immunogenic IRF1+ as well as NFκB+ CB-LCs (Supplementary Fig. 4H, I), SGE did not reduce immunogenic response to *S. aureus*. Together, these results indicate that tick saliva causes tolerogenic polarization of epidermal LCs which remains unchanged in *B. burgdorferi* infection. Associated lack of pro-inflammatory LC response might result in decreased bacterial clearance and lack of memory formation in tick-borne infections.

### Langerhans cells are reduced in *B. burgdorferi* skin infection

Previous studies have described decreased LC numbers in acute cutaneous *B. burgdorferi* skin infection (erythema migrans)[11,29]. However, it remains unknown whether this effect is transient and mediated by the inflammatory response to pathogenic antigens or occurs in response to tick salivary proteins, as suggested by our data. We therefore investigated the frequencies of cutaneous CD207+ LCs in the erythema migrans lesion centers of patients with cutaneous Lyme disease, which form at the site of prior tick bite and compared them to autologous, non-lesional skin (Fig. 5A). While we detected a normal density of LCs in non-lesional skin from a distant skin site of patients with cutaneous Lyme disease, LC percentages were drastically reduced in infected lesional skin (Fig. 5B, C).

### Single-cell RNA sequencing reveals activated, tolerogenic Langerhans cells in *B. burgdorferi* skin infection

To dissect the role of LCs in *B. burgdorferi* skin infection, we performed single-cell RNA-sequencing (scRNAseq) of cutaneous Lyme disease (erythema migrans) skin lesions and non-lesional skin biopsies of the same infected individuals (Fig. 5A, and Supplementary Fig. 5A). Expression of *CD207* in UMAP projection demarcated a clear cluster that was identified as cutaneous LCs by expression of canonical marker genes, including *CD207* and *CD1A* (Fig. 5D). Sub-clustering of LCs revealed differential transcriptional states of LCs isolated from lesional vs. non-lesional skin (Fig. 5E). We next looked at the top significant differentially expressed genes (DEG) between these tissues in an unsupervised fashion (Fig. 5F). Interestingly, lesional skin LCs significantly over-expressed the migration receptor *CXCR4* and the tolerogenic transcription factor *IDO1*. While LCs from *B. burgdorferi*-infected skin expressed higher levels of activation and migration markers, the expression of pro-immunogenic transcription factors was comparable to that of healthy, non-lesional skin LCs (Fig. 5G). Instead, in line with our in vitro experiments, LCs from lesional skin expressed high levels of pro-tolerogenic transcription factors (Fig. 5G). Integration of scRNAseq datasets from erythema migrans skin samples of a previously published cohort[30] confirmed these findings (Supplementary Fig. 5B–D). Together, our data suggest emigration and tolerogenic priming of LCs in patients with tick-borne *B. burgdorferi* skin infection.

### Priming of tolerogenic T cells by Langerhans cells stimulated with tick salivary gland extracts and *B. burgdorferi*

To assess the consequence of tolerogenic LC polarization on induction of an adaptive immune response against tick-borne pathogens, we stimulated moLCs with SGE, *S. aureus* or *B. burgdorferi* and subsequently co-cultivated them with autologous naïve CD4+ T cells. Only moLCs stimulated with *S. aureus* induced follicular helper T cells (Tfh, Fig. 6A), effector helper T cells type-17 (Th17) and type-9 (Th9; Fig. 6B, left and center). The levels of type-2 helper T cells (Th2) did not significantly change after SGE and *B. burgdorferi* stimulation compared to unstimulated controls but decreased after stimulation with *S. aureus* (Fig. 6B, right panel). Importantly, moLCs pre-incubated with SGE induced significantly higher percentages of Tregs, with the same trend overserved after *B. burgdorferi* challenge compared to unstimulated and *S. aureus* treated cells (Fig. 6C), further suggesting the tolerogenic potential of LCs in the context of tick bite and tick-borne pathogen transmission. The numbers of CD69+ tissue resident memory T cells (Trm) did not change significantly, but showed a trend towards an increase when co-cultured with SGE-stimulated moLCs similar to what we previously observed in an experimental tick bite model[5]. On the contrary, CD103+ Trm were only increased after incubation with *S. aureus*-stimulated moLCs (Supplementary Fig. 6C).

We next investigated T cell priming in the integrated erythema migrans skin dataset (Supplementary Fig. 5B, C), including newly generated and published samples[30]. No differential UMAP clustering was observed between T cells isolated from lesional vs. non-lesional

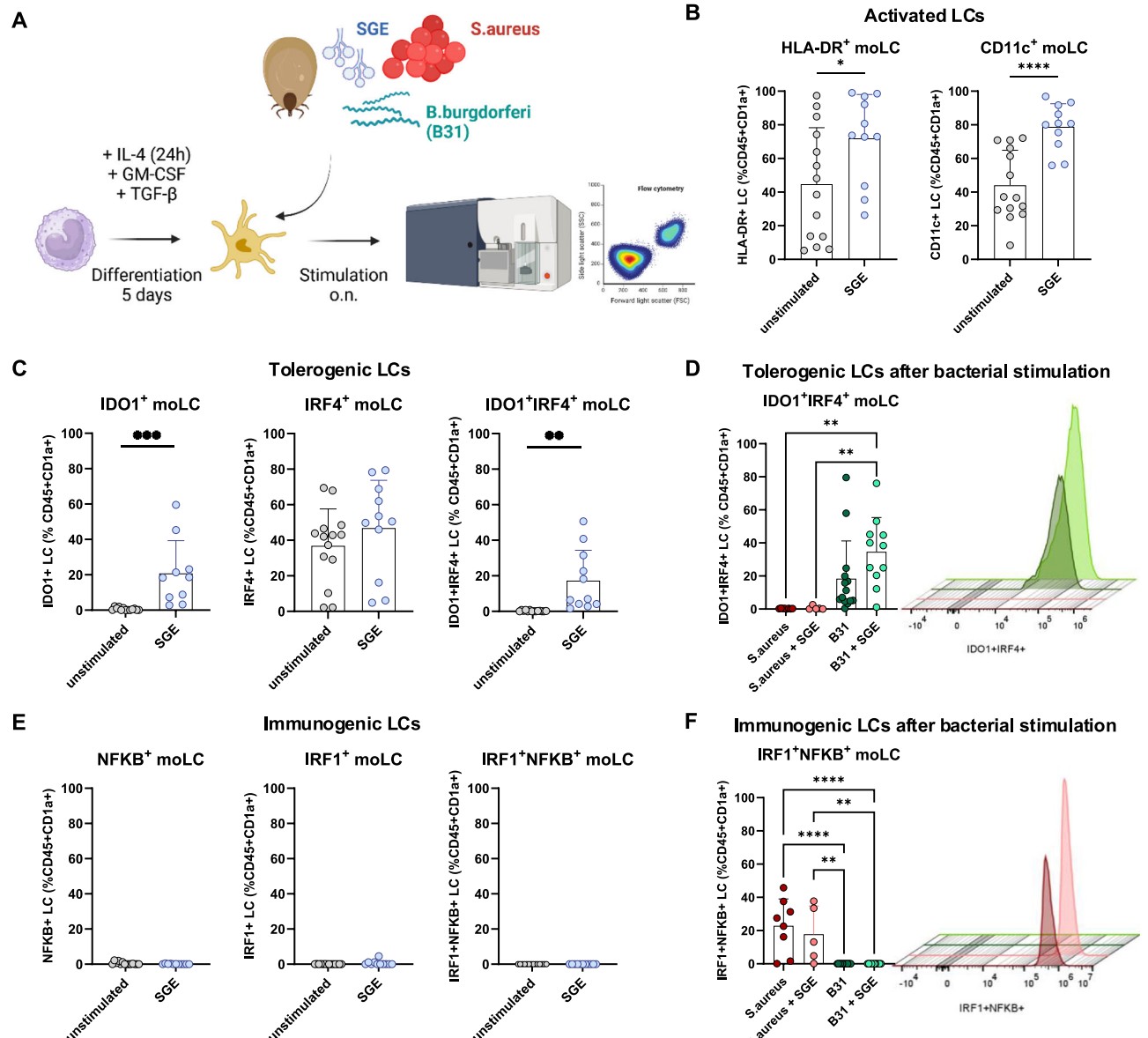

**Fig. 4 | Langerhans cells up-regulate tolerogenic transcription factors after stimulation with tick salivary gland extracts and tick-borne *B. burgdorferi*.**
**A** Conceptual overview of LC differentiation and overnight stimulation with SGE, *S. aureus* and *B. burgdorferi* (B31). Created in BioRender. Stary, G. (2025) https://BioRender.com/n1gbezu. **B** Flow cytometry data of LC activation markers HLA-DR and CD11c of unstimulated (*n* = 14) and SGE-stimulated (*n* = 11) moLCs *P* = 0.036, *P* ≤ 0.0001. **C–E** Expression of the tolerogenic transcription factors IDO1 and IRF4 in unstimulated (*n* = 14) moLCs or after stimulation with SGE (*n* = 11) *P* = 0.007, *P* = 0.0011 (**C**) and/or *S. aureus* (*n* = 5) and *B. burgdorferi* (*n* = 11) *P* = 0.0014,

*P* = 0.0071 (**D**). Immunogenic transcription factor expression (NFκB, IRF1) in unstimulated (*n* = 14) moLCs stimulated with SGE alone (*n* = 11) (**E**) and/or *S. aureus* (5) and *B. burgdorferi* (*n* = 11) *P* ≥ 0.001, *P* = 0.0046, *P* = 9.0065 (**F**). Data are plotted as mean percentage of cells ± SD. *N* represent biological replicates. Statistical significance was calculated using unpaired Student's *t* test (two-tailed) for the comparison of two groups and ordinary one-way ANOVA with Turkey multiple comparison correction for the comparison of more than two groups (*P* < 0.05; **P* < 0.01, ***P* < 0.001; ****P* < 0.0001).

skin (Fig. 6D). However, similar to our findings in SGE-stimulated LC-T cell co-cultures, we observed increased expression of Trm marker genes (*CD69*, *FABP5*) in erythema migrans lesions. Notably, expression of T cell lineage and cytokine genes revealed little differences between groups (Fig. 6E), arguing for a blunted immune response to tick-borne *B. burgdorferi* infection in the skin.

**Tick salivary gland extract-stimulated Langerhans cells induce aberrant T cell responses to *B. burgdorferi* in an ex vivo immune spheroid culture model**
To assess the consequences of tolerogenic LCs migrating into lymph nodes, we investigated the influence of SGE-stimulated LCs

on T cell induction in a lymphoid tissue immune spheroid culture derived from human splenic cells as described by Wagar et al. (Fig. 6F, Supplementary Fig. 6A, panel I and Supplementary Fig. 6B)[31]. SGE-primed autologous spleen-derived moLCs induced markedly lower Tfh and Th9 polarization with a simultaneous induction of Th2 cell differentiation in the immune spheroid culture model, independent of pathogen presence (Fig. 6G, H). In line with our co-culture data, *B. burgdorferi* infection of immune spheroid cultures (Supplementary Fig. 6A, panel ii) induced a Th2 phenotype in naïve T cells, while failing to mount the appropriate Th17 and Th9 responses as well as Trm induction as observed after *S. aureus* infection (Fig. 6I, J and Supplementary Fig. 6D, E). These

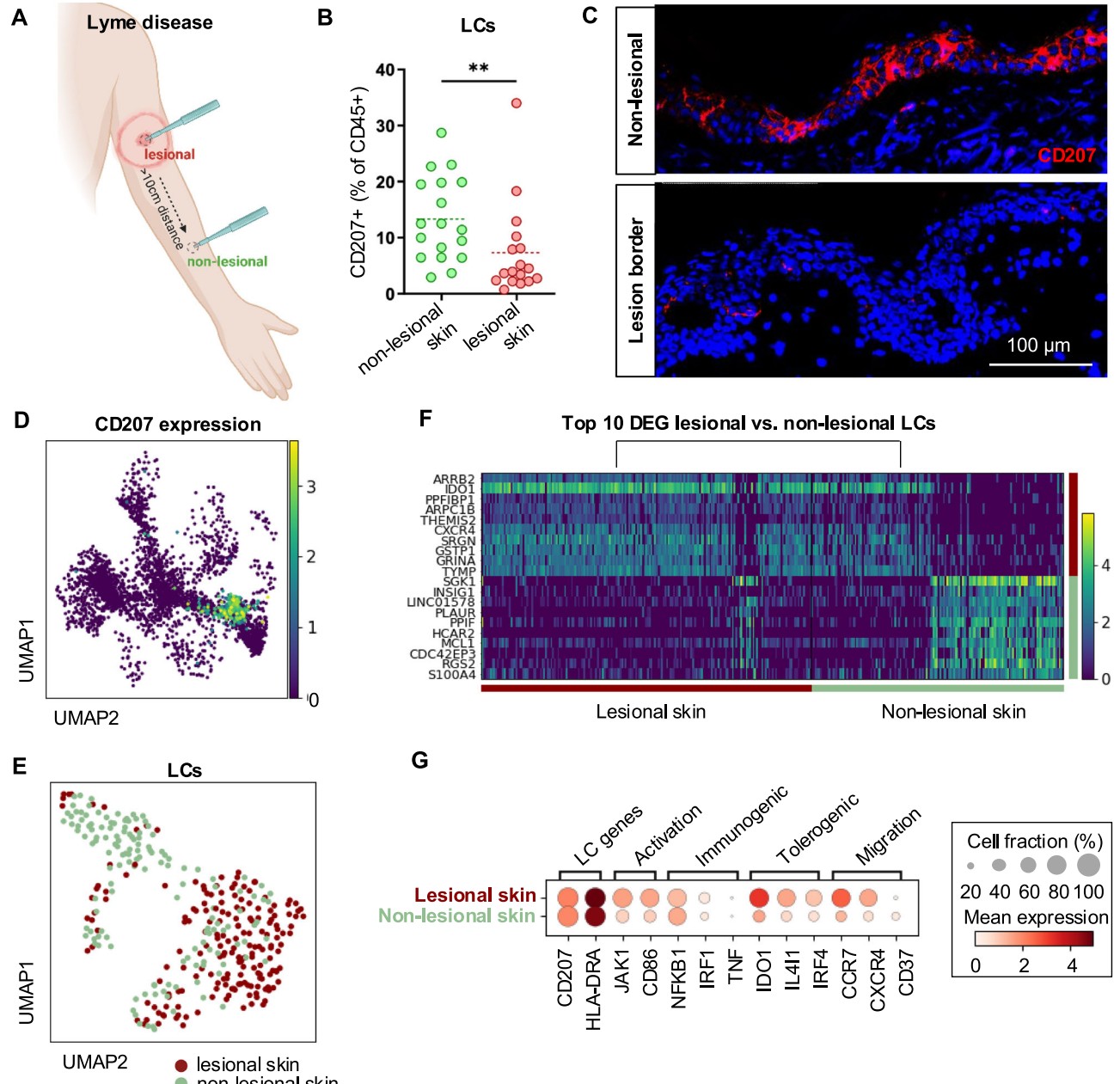

**Fig. 5 | Decreased numbers of Langerhans cells with a tolerogenic phenotype in cutaneous Lyme disease lesions. A** Illustration of lesional and non-lesional skin biopsy sampling sites in patients with cutaneous Lyme disease. Created in BioRender. Strobl, J. (2025) https://BioRender.com/fr04qn3. **B** Flow cytometry data of CD207+ LC in single-cell suspensions obtained from cutaneous Lyme disease patients (n = 18) P = 0.0098. **C** Representative IF images of CD207+ (red) LCs and DAPI (blue) in the epidermis of lesional and non-lesional skin sections. **D** UMAP projection of *CD207* expression in the CD45+ immune cell fraction in scRNAseq data from lesional and non-lesional skin cells of 2 cutaneous Lyme disease patients sampled as shown in (**A**). **E** UMAP of *CD207* and *CD1A*-expressing LC cluster from (**D**). **F** Heatmap for the top 10 differentially expressed genes (DEG) between lesional and non-lesional LCs from (**E**). **G** Dot plot showing the expression of genes associated with LC activation, migration and immunogenic and tolerogenic LC phenotypes in cells from (**E**). Statistical significance in (**B**) was calculated using unpaired Student's *t* test (two-tailed) for the comparison of two groups (n = 18; **P < 0.01).

results support the idea of an altered adaptive effector T cell induction by LCs in response to tick bite and *B. burgdorferi* infection.

## Discussion

Our study identified tick saliva-induced LC activation, migration and pro-tolerogenic polarization as mechanisms of impaired adaptive immune responses upon arthropod feeding and tick-borne pathogen transmission. Using a combination of experimental systems, including complex ex vivo models and single-cell profiling of primary patient samples, we show that human LCs are activated upon contact with tick salivary gland extracts and emigrate from the epidermis to the lymphatics to induce a tolerogenic T cell response.

LCs are instrumental for skin homeostasis, evidenced by complex mechanisms of their (re-)population dynamics from different precursors in embryogenesis and adult life. During embryogenesis, yolk sac and fetal liver monocyte/macrophage-derived LCs seed the epidermis, where they persist throughout life and show self-renewing capacity, maintaining their population by local proliferation. They play a crucial role in immune surveillance and tolerance and regulate immune responses to avoid excessive inflammation, facilitating tissue homeostasis[27,28]. Especially during inflammation or tissue injury, the

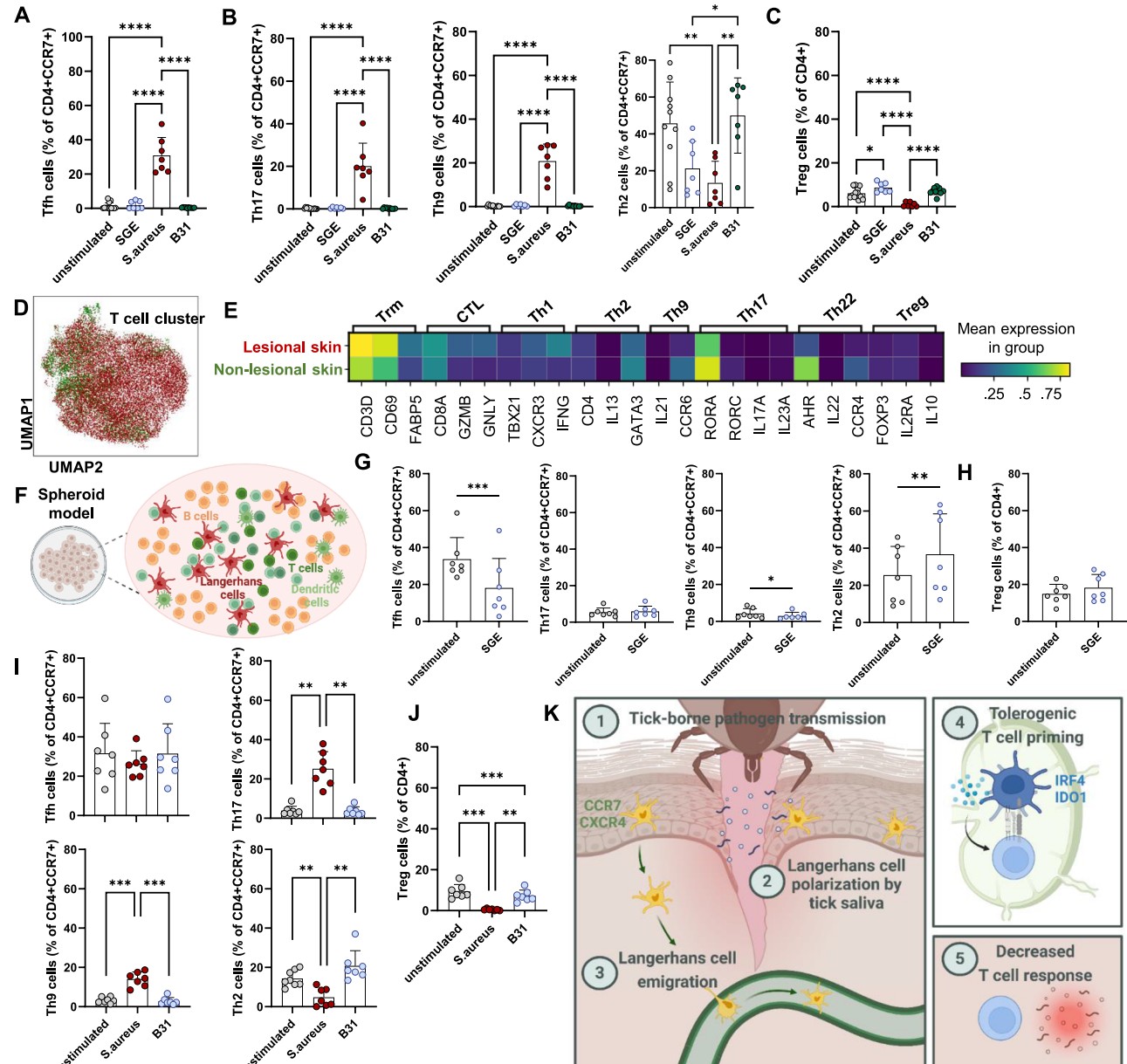

**Fig. 6 | Lack of immunogenic T cell polarization in erythema migrans lesions and upon Langerhans cell stimulation with tick salivary gland extracts and tick-borne *B. burgdorferi*.** Flow cytometry data of Th cell subset induction after co-incubation with SGE-, *S. aureus*- or *B. burgdorferi*-stimulated moLCs (*n* = 7). Data shown as mean percentages +SD of Tfh *P* ≤ 0.0001 (**A**), Th17 *P* ≤ 0.0001 (**B**, left), Th9 *P* ≤ 0.0001 (**B**, center), Th2 *P* = 0.0073, *P* = 0.0337, *P* = 0.0049 (**B**, right panel) among CD4+CCR7+ T cells and **C** Treg among CD4+ T cells *P* ≤ 0.0001. **D** UMAP projection of unsupervised clustering in the T cell-subcluster (*PTPRC*, *CD3E*, *CD3D*-expressing cells) of the integrated erythema migrans (lesional) and healthy control (non-lesional) skin datasets (*n* = 8, Supplementary Fig. 5B). **E** Heat map of canonical Th lineage marker gene expression in cells from (**D**). **F** Schematic of the immune spheroid culture model. Created in BioRender. Strobl, J. (2025) https://BioRender.com/ywj6rqo. **G**, **H** Flow cytometry data of Th cell subset induction in immune spheroid culture model after co-incubation with autologous moLCs stimulated with SGE overnight (*n* = 7) *P* = 0.0008, *P* = 0.0332, *P* = 0.0068. **I**, **J** Flow cytometry

data of Th cell subset induction in the immune spheroid culture model after co-incubation with *S. aureus* or *B. burgdorferi* (*n* = 7) (Th17) *P* = 0.0012, *P* = 0.0014, (Th9) *P* = 0.0004, *P* = 0.0003, (Th2) *P* = 0.0018, *P* = 0.0023, (Treg) *P* = 0.0010, *P* = 0.0006, *P* = 0.0016. **K** Graphical abstract. Tick feeding causes the up-regulation of CCR7 and CXCR4 on human epidermal LCs, which are polarized by tick saliva and emigrate to the lymphatics to mediate immune tolerance. Consequently, tolerogenic T cell priming and lack of effector T cell differentiation results in an inappropriate T cell response to tick-borne pathogens. Created in BioRender. Strobl, J. (2025) https://BioRender.com/m2w9q4r. **G–J** Data shown as mean percentages +SD of CD4+ or CCR7+CD4+ T cells. **A–C**, **G–L** *N* represent biological replicates. Statistical significance was calculated using paired Student's *t* test (two-tailed) for the comparison of two groups and ordinary one-way ANOVA with Turkey multiple comparison correction for the comparison of more than two groups (*P* < 0.05; **P* < 0.01; ***P* < 0.001; ****P* < 0.0001).

epidermis is temporarily replenished by an initial wave of short-term peripheral blood monocyte-derived LCs (moLCs), which are associated with antigen presentation and immune activation. These cells are superseded by a second wave of steady-state precursor-derived long-term LCs once the inflammation is cleared[21,32]. Testing the concept of

LC populations of human skin in vitro, our data provide evidence that contact with tick saliva maintains tolerance in CB-LCs, a model for steady-state epidermal LCs during homeostasis. Concurrently, tick salivary gland extract directly causes upregulation of pro-tolerogenic transcription factors IDO1 and IRF4 in inflammation-associated moLCs.

Consequently, these tolerogenic LCs fail to induce appropriate T cell responses, resulting in Treg and Th2 cell polarization.

In addition, we found that LCs upregulate tolerogenic transcription factors after stimulation with the tick-borne pathogen *B. burgdorferi* in vitro, whereas incubation with *S. aureus*, a pathogen known to cause cutaneous inflammation, resulted in increased expression of immunogenic transcription factors (IRF1, NFκB). This is in line with a recent report from Pan et al., who found immunogenic LC transcription factors and increased Th2 cytokine expression in *S. aureus*-primed LC-T cell co-cultures[33].

LC polarization in an immune spheroid culture model resulted either in tolerance/Th2 induction in case of SGE and *B. burgdorferi* incubation, or Th17 and Th9 effector cell differentiation and induction of an inflammatory immune response following *S. aureus* stimulation. In line with our findings, increased signaling with Th2-cytokines IL-4 and IL-13 was observed in skin of patients after tick bite[34]. In an in vitro mouse study, tick saliva inhibited the production of pro-inflammatory cytokines (IL-12, IL-6 TNF-α) by DCs stimulated with Toll-like receptor-4 ligand, while increasing the release of tolerogenic cytokine IL-10[35].

Moreover, in scRNA-seq data sets of patients with a *B. burgdorferi*-associated skin inflammation we identified increased expression of tolerogenic transcription factors in lesional LCs. Importantly, human *B. burgdorferi* infection does not result in appropriate memory formation and re-infections are common[36-38]. We therefore speculate tick saliva-derived bioactive molecules affecting LC function to be the causative mechanism behind suppression of local and systemic adaptive immunity, thereby preventing sufficient recall responses upon pathogen re-encounter. This is supported by the fact that several tick saliva components, including *I. ricinus* serine protease inhibitor *IrSP1*, *Salp15* and *Iris*, were described to have the potential to impair T effector cell polarization. Effects of tick saliva compounds via manipulation of murine APCs or direct in vitro inhibition were shown on T cell proliferation[39], IL-2 production[40], and Th1[41] and Th17 differentiation[42]. The *Rhipicephalus appendiculatus*-derived tick salivary compound *Japanin* can reprogram DC phenotype by inhibition of maturation and downregulation of the co-stimulatory molecule CD86 as well as decrease production of Th17- and Th1-polarizing cytokines[43]. At the same time, *Japanin* led to enhanced expression of co-inhibitory PD-L1 and IL-10 secretion, suggesting its effectiveness in modulating DCs to induce tolerance. Vesely et al. further demonstrated in Langerin-DTA mice that presence of LCs was required for *Ixodes scapularis*-mediated suppression of Th1 response[44].

The fact that tick saliva possesses T cell-specific immunosuppressive functions warrants further research into the use of tick saliva components as therapeutic agents for inflammatory diseases with exaggerated effector T cell responses. The pro-tolerogenic influence of tick saliva proteins on LC transcription factors could be reinforced to achieve tolerance in inflammatory disease. Furthermore, the discovery that tick saliva molecules enhance the migratory capacity of LCs to lymphatics could significantly impact the development of LC-based vaccines[45]. Specifically, inhibiting tick saliva compounds such as *Ixodes scapularis* salivary protein 15 may enhance adaptive immunity and block spirochete transmission[46]. Another strategy to interfere with tick-host interactions and pathogen transmission is the development of anti-tick vaccines: De la Fuente et al. showed that targeting *subolesin* and its orthologue in insects, *akirin*, effectively reduced tick infestations and pathogen infection[47]. Overall, the emergence of ticks and expansion of their natural habitat warrants development of creative preventive strategies against tick-borne pathogens.

In conclusion, our findings indicate an elevated risk of pathogen transmission from natural tick bites due to reprograming of LCs, which induces tolerance by skewing effector T cell polarization in the draining lymph nodes. These findings reveal the profound impact of arthropods as disease vectors on the human immune response, carrying significant implications for vaccine development.

## Methods

### Patient sampling
Healthy, adult individuals with a recent tick bite in medical history (≤7 days) or erythema migrans were recruited to the study (Supplementary Table 1). After fully informed written consent, punch biopsies of the site of tick bite and healthy skin of a distant body site were taken. In addition, skin punch biopsies were obtained from two individuals with active *B. burgdorferi* skin infection (erythema migrans lesion center and non-lesional skin >10 cm distant to infection site). *B. burgdorferi* skin infection was confirmed using routine PCR, which was positive in both cases for the species *B. afzelii*. Tissue was fresh-frozen for immunofluorescence (IF) work-up or immediately processed by enzymatic digestion for flow cytometry and scRNA-seq. The study was approved by the Ethics Committee of the Medical University of Vienna (ECS 1281/2018) and was performed in accordance with the guidelines of the Declaration of Helsinki.

### Tissue immunofluorescence
Multiplex IF stainings of 6 μm-thickness, acetone-fixed skin cryosections were performed with directly labelled antibodies (Supplemental Table 2) as described before[5]. In brief, skin sections were stained with primary antibody for 2 h or overnight and signal was subsequently enhanced with a goat anti-mouse IgG antibody for 30 min, followed by counterstaining with 4′,6-diamidino-2-phenylindol (DAPI) (Sigma). Specificity of IF staining was controlled with isotype-matched conjugated antibodies. Images were acquired using a Z1 Axio Observer microscope equipped with a LD Plan-Neofluar 20x/0.4 objective (Zeiss) and quantified using TissueQuest image analysis software (TissueGnostics).

### Sampling for ex vivo tick bite model
Patients undergoing skin reduction surgeries (excess skin removal, including abdominoplasty, brachioplasty, thigh lift and body lift) performed at the Department of Plastic, Reconstructive and Aesthetic Surgery, Medical University of Vienna, were recruited to the study. Excised tissue (original anatomical location: abdomen, upper lower extremities) with 10 × 10 cm (full thickness skin and adipose tissue) was explanted and immediately placed in RPMI-1640 medium (Gibco). As described before[5], for the experimental tick bite model 30.5 μl PBS or SGE at a concentration of 20.6 μg/ml, diluted in PBS and supplemented with protease inhibitor cocktail (Merck), were injected approximately subepidermally into skin explants using an insulin syringe. Subepidermal injection was performed in a manner analogous to intradermal skin testing, resulting in a small, well-demarcated intradermal elevation. However, leakage of the injected substance into deeper layers cannot be excluded, similar to distribution of injected saliva into multiple skin layers during a tick bite. Explants injected with SGE or PBS were incubated in antibiotic-free sterile RPMI-1640 medium (Gibco) supplemented with 2% fetal bovine serum (Gibco) for 24 h at 37 °C. Biopsies from the injection sites were taken, snap frozen, cryosectioned, and subjected to tissue IF staining and analysis as described above.

### Sample processing and flow cytometry
Enzymatic tissue dissociation of skin biopsies was performed using 5 IU collagenase IV (Worthington Biochemical) together with DNase I from bovine pancreas (Sigma) and the gentleMACS (Miltenyi Biotec) dissociation program for human skin. Single-cell suspensions were stained with surface and intracellular markers[48], in cases of transcription factors using the Foxp3/Transcription Factor Staining Buffer Set (eBioscience, ThermoFisher). Antibodies are listed in Supplementary Table 3. Single-cell suspensions were acquired using FACS Diva software on a FACS Aria III cell sorter (BD Biosciences) or SpectroFlo software on an Aurora spectral analyzer (Cytek). Flow cytometry data were analyzed using FlowJo software (Version 10, BD Biosciences).

### Langerhans cell emigration/isolation from epidermal sheets

To isolate human epidermal LCs, 8 mm biopsies were taken from skin reduction surgeries as described above. Epidermis was removed from adjacent dermal tissue using a scalpel and subsequently digested overnight at 4 °C in Dispase II, 2.2 μ/ml (Roche). The next day epidermal sheets were separated from the rest of the tissue using a forceps and cultured for 24 h in RPMI-1640 medium (Gibco) supplemented with 10% FCS (Gibco) and 1% penicillin-streptomycin (Gibco), containing 5 μg/ml SGE or PBS. Number of emigrated LCs (CD45⁺CD207⁺CD1a⁺ cells) in the supernatant after 24 h were assessed using flow cytometry (Supplementary Fig. 1H). Epidermal sheets were stained by incubation with fluorescently labelled CD207 antibody, and nuclei were counterstained with DAPI before acquisition at a confocal laser scanning microscope (Olympus IX38).

### In vitro monocyte-derived Langerhans cell differentiation and stimulation

Monocytes were isolated from healthy donor PBMCs using CD14⁺ magnetic microbeads (Miltenyi) and further differentiated into LCs using RPMI-1640 medium (Gibco) supplemented with 10% FCS (Gibco), 1% Glutamax (Gibco), 5 μM β-mercaptoethanol (Gibco), 1% penicillin-streptomycin (Gibco), 100 ng/ml rh-GM-CSF (Immuno-Tools), 10 ng/ml rh-IL-4 (Peprotech), and 10 ng/ml rh-TGF-β (Peprotech). After 24 h and 3 days, fresh media without IL-4 was added. At day 5, moLC differentiation efficiency was assessed by flow cytometry and cells were stimulated with 40 μg/ml human saliva or SGE for 48 h. Subsequently, activation and migration marker expression were assessed by flow cytometry (described above), and collagen invasion assay was performed.

### In vitro generation of CD34⁺ progenitor cell-derived Langerhans cells

Isolation of CD34⁺ hematopoietic progenitor cells and LC differentiation was performed following a published protocol[22,49]: In detail, CD34⁺ cell isolation was performed by magnetic sorting with the EasySep human CD34 positive selection kit (StemCell) according to manufacturer's instructions. Umbilical cord blood was obtained from healthy donors during full-term delivery within the ethical approval (EK 26-520 ex 13/14) obtained from the Medical University of Graz. CD34⁺ cord blood hematopoietic progenitor cells were expanded for 3 days in serum-free X-VIVO 15 medium (Lonza) supplemented with 50 ng/mL stem cell factor (SCF) (Peprotech), 50 ng/mL Fms-related tyrosine kinase 3 ligand (FLT3-L) (Peprotech) and 50 ng/mL thrombopoietin (Peprotech). Progenitor LCs were generated by using 8 × 10³ pre-expanded CD34⁺ cells cultured in 96-well tissue culture plates in serum-free CellGro GMP DC medium (CellGenix) supplemented with 100 ng/mL GM-CSF (Peprotech), 2.5 ng/mL TNFα (Peprotech), 50 ng/mL FLT3-L (Peprotech), 20 ng/mL SCF (Peprotech), and 1 ng/mL TGF-β1 (Peprotech). Purity of isolated fractions was analysed by flow cytometry.

### Tick and human saliva preparation

*Ixodes ricinus* salivary gland extracts were obtained from the Slovak Academy of Science (Bratislava, Slovakia)[5]. Specific pathogen free adult *I. ricinus* laboratory colony ticks were allowed to feed on rabbits for 5 days, after which their salivary glands were dissected, pooled in ice-cold PBS and shipped and stored on −80 °C. Once received and thawed salivary glands were homogenized, and SGE was used for experiments. Human saliva was collected from healthy individuals, diluted with PBS and filtered through a 0.3 μm pore-size filter. Human saliva mix of 5 donors was used for experiments. Saliva protein concentrations were measured by Nanodrop (Thermo Fisher).

### Collagen gel invasion assay

Migration chambers were custom made by assembling a petri dish containing a 17 mm hole in the center with two class coverslips using paraffin. 3D collagen scaffold consisting of 1.67 mg/ml bovine collagen I (Advanced Biomatrix) supplemented with 0.6 mg/ml rh-CCL19 (ImmunoTools) and buffered to physiological pH with Minimum Essential Medium (Gibco) and sodium bicarbonate (Sigma) in a 1:2 ratio, was added to the migration chambers. After polarization of the collagen matrix at 37 °C for 30 min, moLCs stimulated with 40 μg/ml human saliva or SGE were added on top and allowed to invade into the scaffold. Images were taken after 48 h with an Olympus IX53 microscope, equipped with an Olympus TH4-200 illumination source and a Hamamatsu Orca Spark camera.

### Bacterial stimulation experiments

We co-incubated moLCs with *S. aureus* (wild-type methicillin-resistant *S. aureus* strain USA300 LAC, kindly provided by Fabio Bagnoli (GSK Vaccines, Siena, Italy)) or *B. burgdorferi* (strain B31) cultured in bacterial growth phase, with and without SGE overnight. Bacteria and SGE were mixed and subsequently the mix of *B. burgdorferi* and SGE was directly added to the cells. SGE was used at 50 μg/mL dilution in RPMI-1640 medium + 10% FCS (Gibco). 1 × 10⁸ CFU of *S. aureus* were cultured for 48 h at 37 °C and 120 rpm agitation in LB Broth medium (Invitrogen, Thermo Fisher) and used for bacterial stimulation experiments during the exponential growth phase at optical density (OD) 3.5. B31 was cultured at 34 °C in a modified BSK II medium containing of gelatin solution and CMRL-1066 media (Sigma Aldrich), supplemented with Neopeptone (Gibco), yeast extract, HEPES sodium salt (Sigma Aldrich), glucose, sodium bicarbonate, sodium citrate dehydrate, sodium pyruvate, N-acetyl-d-glucosamine, 30% bovine serum albumin solution (Sigma Aldrich) and heat-inactivated rabbit serum (R&D systems)[5,50]. For bacterial stimulation experiments 1 × 10⁶ borrelia were used per 100,000 cells.

### Langerhans cell – T cell coculture

Monocytes and T cells were isolated from healthy donor PBMCs using CD14 Microbeads (Miltenyi) and the naïve CD4+ T cell isolation kit (Miltenyi), respectively. Monocytes were further differentiated into LCs as described above, and naïve T cells were cultured in ImmunoCult™-XF T Cell Expansion Medium (StemCell) supplemented with rh-IL-2 (200 IU; Peprotech). MoLC differentiation efficiency was in the range of 50–80% for all experiments. On day 5, moLCs were incubated with SGE, *S. aureus* or *B. burgdorferi* overnight as described before. Subsequently, moLCs were washed 1× with PBS (Gibco) to remove bacteria and SGE and co-cultivated with autologous naïve CD4+ T cells for 6 days in RPMI-1640 medium (Gibco) supplemented with 10% FCS (Gibco) and 1% penicillin-streptomycin (Gibco). Subsequently, polarized T cell subsets were analyzed by flow cytometry as described above.

### Human immune spheroid culture model

Spleen tissues were obtained from organ transplant donors (Ethics Committee protocol number 2166/2019). Single cell suspension preparation from spleen tissue and subsequent spheroid cultures were performed as described before[31,51]. In short, single cell suspensions gained from donor spleens were seeded in Ultra-Low Attachment 96-well plates (Corning) at 1.5 × 10⁶ cells per well and cultured in RPMI-1640 medium (Gibco) supplemented with 10% fetal calf serum (FCS; Gibco), 1% MEM Non-Essential Amino Acids Solution (100X; Gibco), 1 mM sodium pyruvate, 1 × insulin-transferrin-selenium (Gibco) and 0.5 μg/mL recombinant human B lymphocyte stimulator (BAFF; Immunotools, Friesoythe, Germany). Simultaneously, organoids were co-incubated with live *B. burgdorferi* (B31) or *S. aureus* bacteria in active growth phase as described above. After 48 h, 100 μ/mL penicillin-streptomycin (Gibco) and 100 μg/mL normocin (Invivogen, San Diego, CA, USA) were added to the culture medium to prevent bacterial overgrowth. For APC co-culture, moLCs were isolated from autologous spleen donors with the CD14⁺ cell isolation kit (Miltenyi) as described above and added to the immune spheroid culture model after

overnight pre-treatment with SGE. Cell culture medium was replaced every second day and spheroid cultures were maintained at 37 °C for 7 days during which the single cell suspensions arranged themselves into lymphoid organ structures as described by Wagar et al. After 7 days, the immune spheroid culture model was harvested and single-cells analyzed by flow cytometry.

## Statistics & reproducibility

Statistical analyses of immunofluorescence and flow cytometry data were performed using GraphPad Prism 10.2.1 (GraphPad Software). Statistical significance was determined by Student's *t*-test when comparing two groups and (repeated-measures) two-way analysis of variance (ANOVA), or mixed-effects analysis when comparing three or more groups. Tukey or Dunnett multiple comparison post-test was used for multiple testing correction. Significance was set at a *p*-value of less than 0.05. No statistical method was used to predetermine sample size. Sample sizes were chosen based on feasibility of obtaining primary human skin samples and patient material. No data were excluded from the analyses. Number of replicates, the specific statistical tests applied to each dataset, including corrections for multiple comparisons where appropriate, are described in the corresponding figure legends.

## Single cell RNA sequencing and bioinformatic analyses

For droplet-based scRNA-seq, CD45+ live cells were sorted into PBS/ 10% FCS for library preparation with the Single-Cell 5' Library & Gel Bead Kit v2 (10X Genomics) according to the manufacturer's instructions. As described before[52], sorted cells were partitioned into gel bead-in-emulsions (GEM) for cDNA synthesis and amplification. The samples were sequenced on a NovaSeq SP platform (Illumina, 50-bp paired-end configuration). Raw sequencing data was processed using the Cell Ranger Single-Cell Software Suite (version 3.1.0, 10× Genomics Inc) and aligned to the GRCh38 reference genome. Filtering steps were applied for cells (detected gene number <200, mitochondrial gene expression <20%) and genes (expression in >3 cells). As samples from two donors were not processed at the same date, batch correction was performed using BBKNN graph-based data integration algorithm[53]. Public scRNA-seq datasets of cells isolated from erythema migrans skin lesions (*n* = 8) were obtained from Jiang et al.[30] via the gene expression omnibus (GEO) database, accession number GSE169440[54]. Our newly generated datasets (*n* = 2) were integrated using the pre-processing parameters described by Jiang et al., and objects were merged, followed by batch correction using the BBKNN algorithm. Differential gene expression analysis and data visualization was performed with the SCANPY toolkit[52,55].

## Reporting summary

Further information on research design is available in the Nature Portfolio Reporting Summary linked to this article.

# Data availability

scRNA-seq datasets related to this article are available via NCBI's Gene Expression Omnibus (GEO) database in form of de-identified count matrices, accessible via: https://www.ncbi.nlm.nih.gov/geo/query/acc. cgi?acc=GSE306532, or upon request from the corresponding first author.

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

## Acknowledgements

This work was supported by the Medical Scientific Fund of the City of Vienna (JS, 19055), the La Roche-Posay Research Awards Europe (JS), ESCMID Research Grant (JS), the LEO Foundation (GS, LF-OC-21-000806) and the Austrian Science Fund (GS, FWF PAT 8019123). The authors thank Kveta Brazdilova for technical support in sequencing data upload. Graphics created with BioRender as detailed in the figure legends corresponding to first appearance of the graphic.

## Author contributions

J.S.: Conceptualization, funding acquisition, methodology, investigation, data analysis, writing (original draft); L.K.: methodology, investigation, data analysis, writing (original draft); J.E., S.W., L.M.G., S.C., A.K., T.F., P.B.: methodology, investigation; J.S., C.F., M.M., K.S., G.S.: Patient recruitment, clinical assessment and sampling; M.W., K.S., M.M., H.Str, H.Sto, G.S.: Conceptualization, funding, supervision; All authors: writing (review & editing).

## Competing interests

J.S. discloses serving as consultant for Pfizer (C4601003 pfVLA15a Adjudication Committee). M.M. is an unpaid member of the Executive Committee of ESGBOR, the ESCMID Study Group for Lyme Borreliosis and discloses serving as consultant for Pfizer unrelated to this study. The other authors have declared that no conflict of interest exists.
