## [Transparent Peer Review file · Nature Communications]

Human epidermal Langerhans cells induce tolerance and hamper T cell function upon tick-borne pathogen transmission

Corresponding Author: Professor Georg Stary

Version 0:

Reviewer comments:

Reviewer #1

(Remarks to the Author)

Reviewer comments on Strobl et al.: Human epidermal Langerhans cells induce tolerance and drive immune evasion upon tick-borne pathogen transmission

The manuscript by Strobl et al. describes the relationship between the immune system in the host skin, skin-associated pathogenic bacteria (both transmitted and non-transmitted by ticks) and components of tick saliva, with a focus on dendritic cells and their tolerogenic potential. Using a variety of complex in vitro and ex vivo human model systems, the authors provide new insights into how tick feeding dampen an effective immune response against potential tick-borne infection. More importantly, the tolerogenic phenotype of Langerhans cells is even more pronounced in the presence of tick-borne *Borrelia burgdorferi* and the spirochetes themselves are shown to drive the responses in favour of anti-inflammatory Th2 and Treg responses. I consider such research very important, as we need to search for the reasons why *Borreliae* cause long-lasting infections and why the primary infection does not provide protective immunity.

I raise several questions and comments about the manuscript and the results presented:

- 1) Throughout the manuscript, the abbreviation TS is used instead of "tick saliva". In fact, instead of saliva (=salivation product), salivary gland extract (SGE) was used, which is indeed mentioned in the Methods, but its composition is not the same as that of saliva and the use of the term "saliva" in this context is, in my opinion, confusing.
- 2) For what reason was human saliva used as controls in Figure 3? Please comment more on why CB-LCs tend to respond to HS.
- 3) What was the rationale in using the B31 strain of *B. burgdorferi*? Given that B31 is not transmitted by *Ixodes ricinus* in nature, would you expect a different scenario when using a European strain of *Borrelia* (e.g. *Borrelia afzelii* CB43)?
- 4) In experiments where the cells were treated with both salivary gland extract and *Borrelia* spirochetes, were these two treatments performed simultaneously or was there any pre-incubation step?
- 5) Lines 301-303: In addition to *Amblyomma cajennense* saliva, is there any published evidence regarding the role of *I. ricinus* saliva on cytokines and/or T cell polarization?
- 6) Lines 396 and 411: The authors report that the efficiency of moLC differentiation and the purity of isolated fractions were analysed by flow cytometry. What was the resulting efficiency/purity of these cells?

Formal points:

Line 134 – reference to Fig. 3F is missing in the text.

Line 440 – replace PBMCS with PBMCs.

Reviewer #2

(Remarks to the Author)

This excellent manuscript from Strobl, et al, describes an intriguing mechanism whereby langerhans cells are phenotypically altered by tick saliva during *Borrelia burgdorferi* infection, which promotes immune tolerance. The key results are supported by robust data, which was generated using multiple complementary methods. These included analysis of human patient

samples, validation with an ex vivo human skin model, imaging, flow cytometry, functional assays, in vitro cell coculture, and transcriptomics analysis by scRNA-seq. Overall, the conclusions were highly significant and were supported by appropriate analysis of the experimental data. However, there were a few minor concerns with the manuscript, as outlined below:

1. Figure 1: The experimental Tick Bite experiment seems to be slightly underpowered. In Fig 1D, the % CD1a+ cells were modestly significant, and, more importantly, there appears to be a nonsignificant trend in the CD1a+ cells/mm², which is the most relevant measurement. Increasing the N of this ex vivo experiment will likely result in achieving statistical significance, but a power analysis and additional replicates should be performed to confirm this. It appears the data in Fig 2A-F may be from the same experiment. If so, additional replicates may also improve statistical power for these data.
2. Figure 2: In Fig. 2G-H, please re-analyze the IF data to provide a geometric mean distance between LC and lymph vessels, rather than using a "close to" cutoff of 20um distance. Additionally, the title of Fig 2H should be changed to reflect the data (all LC are not "in" lymph vessels).
3. Fig 4 & 6. Please provide the multiplicity of infection (MOI) for the co-culture experiments involving stimulation with Bb and Sa, in the figure, text, and methods. Also, these experiments are not infections, which involve an in vivo model. Please change this wording to stimulation or coculture in the figure and text.
4. Fig 6: the ex vivo lymphoid organoid model is not adequately explained in the manuscript text or figure 6. Please include a brief description of the model, currently provided in Supplementary Fig 6, in the main body of the text/figure to provide sufficient information needed to understand the methodology and interpret results.
5. Overuse of abbreviations made reading the manuscript somewhat burdensome, particularly overuse of TB/TS. Please revise to spell out "tick bite" and "tick saliva" throughout the manuscript, figures, and figure legends.
6. While use of LC is appropriate in the manuscript body, abbreviations should be avoided in the graphical abstract.
7. Similarly, abbreviations should be avoided, if possible, in figure titles and section/subsection headings.

Reviewer #3

(Remarks to the Author)

The manuscript titled "Human epidermal Langerhans cells induce tolerance and drive immune evasion upon tick-borne pathogen transmission" by Strobl et al. addresses the topic of how tick saliva (TS) and tick-borne pathogens may reprogram human Langerhans cells (LCs) to promote immune tolerance. The authors present a range of experimental models, including ex vivo systems and lymphoid tissue organoids, to examine LC migration, chemokine receptor expression, and T cell polarization. They report that TS induces upregulation of IDO1 and IRF4, which are associated with tolerogenic immune responses, while also promoting a shift toward Th2 and regulatory T cell (Treg) polarization. However, while the study provides some insights, several claims are not fully supported by the presented data, critical controls are lacking, and some conclusions require additional clarification. Below are my specific comments for consideration:

Major comments:

1. Title: The title should be rewritten to tone down its claims. The current wording "... and drive immune evasion" implies a level of direct evidence that is not fully supported by the data presented.
2. Introduction: On line 54, it is important to note that skin Langerhans cells (LCs) are not derived solely from monocytes. They also originate from embryonic precursors. Please revise this section to provide an accurate description of the origin of LCs.
3. Figure 1: Line 77: The authors found a statistically significant reduction in CD1a+ cells in the epidermis of the TB group. However, the results for CD207+ cells were not statistically different. Given that both CD1a and CD207 are markers for LCs, the authors should explain this discrepancy.
4. Figure 1: Additional staining: Did the authors stain for other antigen-presenting cells (APCs) in the epidermis? It is possible that other APCs, such as inflammatory LCs with lower expression of CD207, may replace steady-state LCs. These cells could have a significant impact on the immune response. The authors are encouraged to stain for HLA, which could provide insights into the presence of other APCs that might compensate for the loss of steady-state LCs.
5. Figure 1: Clarity issue: The sentence "...This effect was independent of the sampled body site..." is unclear. The authors did not present a comparison of HC and TB at different body sites, making this claim difficult to interpret. Please clarify this statement or provide supporting data.
6. Figure 1D: In the ex vivo model, the injection of TS leads to an overall reduction in the frequency of CD1a+ cells. However, when calculated as a percentage per mm², the reduction is not statistically significant. The authors should explain this apparent inconsistency.
7. Figure 2A: Similar to comment 6, here the statistical difference is found only when calculating the percentage per mm², but not the total percentages of CD1a. The authors should provide an explanation for this observation.
8. Figure 2C: In contrast to the epidermis, dermal LCs show no significant expression of CXCR4 and CCR7 between the HC and TB groups. While this is reasonable for CXCR4, CCR7 expression should still be higher on dermal LCs as they require this expression to migrate to the lymph node. This is particularly relevant given that CCR7 expression is significantly higher in the TB group in the epidermis. The authors should address this discrepancy.
9. Figure 2E: In the ex vivo system, CCR7 is not significantly upregulated in either epidermal or dermal LCs. This weakens the authors' claim regarding the presence of LCs in the lymphatics. Can the authors provide flow cytometry plots showing

CCR7 expression on LCs in the epidermis and dermis?

10. Figure 2G: The staining for Podoplanin and CD207 is not convincing. Higher magnification is required to demonstrate that LCs are located in the lymphatics. The supplementary data in Figures S3A-B also show no significant differences in CCR7 expression. The authors must provide flow cytometry data since it is difficult to draw conclusions from the immunofluorescence staining alone.

11. Figure 3: Although the representative FACS plots in panel B display an upregulation of chemokine receptors in the TS groups compared to HS, there were no significant differences between HS and TS in upregulating CXCR4 and CXCR7 on moLCs. Additionally, no differences were observed in the migration assay for these treatments (Figure 3C-E). The same holds true for the experiments involving BC-LC in Figure 3F-G. This undermines the conclusion that TS causes LC migration, as HS appears to have a similar effect on migratory capacity.

12. Figure 4: The important control group of HS used in the previous experiments is missing in this set of experiments. Please add it.

13. Figure 4C: The authors mentioned in the text that IRF4 and IDO1 are transcription factors associated with a tolerogenic setting. However, only IDO1 is upregulated, while IRF4 is not. The authors should clarify the implication of this finding. Additionally, the inclusion of the HS control is important to determine whether the upregulation of IDO1 is TS-specific.

14. Figure 4D: Please also show the levels of IRF4 without combining them with IDO1. Additionally, the co-stimulation with TS+B31 does not significantly increase the expression of IDO1 and IRF4 compared to B31 stimulation alone. Therefore, the statement in lines 182-183, "...an effect that was further increased upon combined stimulation with TS," is incorrect and should be revised.

15. Figure 4F: It appears that TS has no significant effect, as it also fails to reduce the immunogenic response to *S. aureus*. The authors should address this point.

16. Lines 191-193: The data does not justify the conclusions presented in these lines. The authors should revise or provide additional supporting data.

17. Figure 5D: Does the UMAP describing CD207 expression represent the lesional skin sample, the non-lesional skin sample, or an integration of both? The authors should clarify this.

18. Figure 5E: The number of LCs in the UMAP is quite low, particularly in the lesional skin samples. This raises concerns about the reliability of the analysis presented in this figure. The authors should perform random downsampling to demonstrate that the results are not solely due to the lower LC numbers in the lesional group.

Minor comments

1. Line 53: Please add the abbreviation "DC" here before using it in line 63.

2. Fig. 5: Please use the abbreviation "LC" as in the rest of the figures

Reviewer #4

(Remarks to the Author)

In the current study, Strobl J, and Kleissl L, et al., investigated TS-induced LC activation using human primary samples and multiple experimental models.

They have conducted multiple immunostaining analyses using human samples to measure the LC activation and their migration to the dermal layer and lymph vessels. The hypothesis was further assessed using the ex vivo model, however there are some questions about the reliability of data using current model. Additionally single-cell profiling of primary patient samples was performed to understand the tolerogenic T cell response after TS-induced LC activation. The data are original and novel, and the manuscript is well-written, though some key experimental design details are lacking.

The concerns regarding the study are outlined below.

1- Human lymphoid tissue organoid model: while there is a typo in the writing, the authors reported using 150000 cells to generate organoids. First this initial cell number seems excessive, raising questions about generating stable aggregation without centrifugation within just 48 hours. Second such model cannot be named as "organoid", since only one cell type present and the current model does not have the complexity of recently published immune system organoids. Spheroid or aggregate are better representatives of this model. Authors referred to the protocol that published in "Nature Medicine volume 27, pages 125–135 (2021)" to generate their model, but there are key differences between these models that should be provided in detail.

2- In their model, the authors have used "100 U/mL penicillin-streptomycin (Gibco) and 100 µg/mL normocin" to the culture medium and then co-incubation with live *B. burgdorferi* (B31) or *S. aureus* bacteria was performed. These antibiotics are not the first line of therapy for Lyme, but penicillin, and normocin has shown effectiveness against Lyme disease, thus incubation with penicillin-streptomycin and normocin could compromise the data.

3- This reviewer acknowledges that investigators have published their ex vivo model previously, but this skin model is not feasible, and the result should be removed as no conclusion could be made based on this model.

4- The investigators mentioned another model "epidermal sheets" in page 4, and mentioned that epidermal sheets were cultivated with TS-supplemented media, where is the detail of this test? What was the concentration? Which cell and from what source they were supplied? Is it primary epidermal cells or cell line? Media? How could they maintain the LC in culture? Which passage number? Many details are missing here.

5- The abstract should be revised, currently it read like a review paper and aims, methodology and results of the study are unclear.

6- Figure legends are not appropriate, for example what SHC means in Figure 1?

7- Schematics in figure 1A and 1C are the same for skin, based on the text those are representing 2 different conditions, this should be revised. Same applies to schematics in Supp Fig 1.

8- Supp.Fig.1 part C the n number should be 8 for each group and not 18, revise accordingly.

9- Suppl. Fig. 1F: how did investigators measure the number of LCs in supernatant?

- 10- The detail of ex vivo human skin TB model should be provided.
- 11- What was the injection volume for delivering the TS into the skin?
- 12- Page 4 and Figure 1: In the ex vivo culture of human skin culture, blood cells, are typically removed. Even if the authors did not deplete the blood cells, the number of CD45+ and other blood cells would likely be minimal, raising concerns about the accuracy of cell count. Can the authors clarify how they addressed this limitation?
- 13- Page 4 and Figure 1: Another limitation is the limited migration of blood cells (CD45+) from ex vivo skin to the supernatant due to the absence of circulation. Even if those CD45+ cells are migrating from ex vivo skin to the supernatant, the majority are likely originating from the skin edges which are not exposed to the TS. This has potential impact on the results, can the authors address this limitation?
- 14- What was the number of CD207+ cells in the supernatant where no PBS and TS was injected into the skin? What was the number of CD1A+ cells in the supernatant? The investigators used different types of antibodies for same study.
- 15- Figure 2G: The merged figures should be provided.
- 16- Figure 2G: according to the text, the CD207+ cells should be next to the Podoplanin+ cells but these two antibodies are overlapped, and seems the exact cells are positive for the same markers, is there any explanation for this? Otherwise, high quality, high magnification images should be provided confirming the proximity of cells without marker overlap. This also may change the conclusions made in part H.

Version 1:

Reviewer comments:

Reviewer #1

(Remarks to the Author)

The authors have addressed all my questions and recommendations raised in the first review. I wish the authors good luck in their future research.

Reviewer #2

(Remarks to the Author)

The reviewer sincerely appreciated the thorough rebuttal of all points raised in response to the initial manuscript review. The new data and revised text have strengthened the manuscript. No additional comments or concerns.

Reviewer #3

(Remarks to the Author)

I have no further comments; the authors have addressed my concerns

Reviewer #4

(Remarks to the Author)

Authors have addressed some this reviewer's concerns, however the following concerns remained:

- In response to the previous concern "such a model cannot be accurately termed an "organoid," since only one cell type is present, and the current model lacks the complexity of recently published immune system organoids. "Spheroid" or "aggregate" would be better terms to describe this model", authors change the "organoid model" to "immune organoid culture model" which is not scientifically correct. Therefore, this term should be revised throughout the manuscript to a more scientific term for their model and called as "aggregate" or "spheroid".

- Some key experimental details are lacking. For example, they claimed that "We added a paragraph to the methods section describing the epidermal sheets in detail (page 22)." However, there are many details missing here that concerns reproducibility of data. I assume they refer to Page 21. When were LCs emerged? What do authors specifically mean by skin reduction surgeries? What was the anatomical location for those biopsies? Your heading refers to isolation of LCs from epidermal sheets, how many days after culture could you harvest enough LCs? Is it 24h? What was the number of LCs that could be detected in the supernatant from 8 mm biopsies and just after 24h?

- Page 20: the authors reported that SGE were injected "subepidermal" into skin explants using an insulin syringe. However, it's difficult to claim that the injection was strictly limited to "subepidermal" and there is leakage into the dermis and potentially hypodermis.

Page 7, and Figure 2: Authors investigated spatial relations of lymph vessel endothelial cells and LCs in the dermis of tick bite-affected skin. They conducted immunostaining for CD207 (Langerhans cells marker) and podoplanin (lymphatic endothelial cell marker), and then the percentage of LCs in proximity (distance $\leq 20 \mu\text{m}$) and within lymph vessels was measured. However, the images provided in Figure 2G (that supporting the data presented in Fig 2G) and Supplemental Figure 3 do not differentiate two different cell types, the image clearly indicate that the same cells were stained for both CD207 and podoplanin markers, and antibody was not specific. Therefore, the claim should be further validated.

Version 2:

Reviewer comments:

Reviewer #4

(Remarks to the Author)

I have no further comment. Congratulations on your great research.

Point-by point response to reviewers

Reviewer #1

The manuscript by Strobl et al. describes the relationship between the immune system in the host skin, skin-associated pathogenic bacteria (both transmitted and non-transmitted by ticks) and components of tick saliva, with a focus on dendritic cells and their tolerogenic potential. Using a variety of complex in vitro and ex vivo human model systems, the authors provide new insights into how tick feeding dampen an effective immune response against potential tick-borne infection. More importantly, the tolerogenic phenotype of Langerhans cells is even more pronounced in the presence of tick-borne *Borrelia burgdorferi* and the spirochetes themselves are shown to drive the responses in favour of anti-inflammatory Th2 and Treg responses. I consider such research very important, as we need to search for the reasons why *Borreliae* cause long-lasting infections and why the primary infection does not provide protective immunity.

We thank the reviewer for the thorough evaluation of our manuscript and appreciation for the importance of the topic of investigation.

I raise several questions and comments about the manuscript and the results presented:

1) Throughout the manuscript, the abbreviation TS is used instead of "tick saliva." In fact, instead of saliva (= salivation product), salivary gland extract (SGE) was used, which is indeed mentioned in the Methods, but its composition is not the same as that of saliva and the use of the term "saliva" in this context is, in my opinion, confusing.

We agree that the wording is confusing and might omit the fact that tick saliva and salivary gland extract composition differs. Therefore, we now use unabbreviated "tick saliva" and have adapted mentions of "TS" to "SGE" wherever applicable.

2) For what reason was human saliva used as controls in Figure 3? Please comment more on why CB-LCs tend to respond to HS.

In addition to media controls, we aimed at including a biological control to determine whether the effect of saliva was tick-specific or could also be induced by mammalian saliva.

A response of CB-LCs to human saliva was somewhat expected, as we used allogenic human saliva (pool of several saliva donors, different from cord blood donor). However, we find the strong differences observed between HS and TS response quite intriguing and think that this is an important aspect of the manuscript.

We have now performed additional experiments describing differences between tick and human saliva responses in more details. We now report that the expression of HLA-DR, CCR7, CXCR4 and tolerogenic transcription factors is significantly increased in LCs upon tick saliva stimulation alone. Those data are now shown in Supplementary Figure 4, D-E.

3) What was the rationale in using the B31 strain of *B. burgdorferi*? Given that B31 is not transmitted by *Ixodes ricinus* in nature, would you expect a different scenario when using a European strain of *Borrelia* (e.g. *Borrelia afzelii* CB43)?

We selected the *B. burgdorferi* B31 strain primarily due to its well-characterized nature and ease of handling in our laboratory setting. Importantly, our skin infection model was originally established and validated using this strain (Strobl et al., *J Clin Invest*, 2022, DOI: 10.1172/JCI161188), allowing for consistent and reproducible results. However, we acknowledge the reviewer's point regarding geographic relevance. In earlier experiments leading up to our study published in *JCI* in 2022, we also employed the European *Borrelia afzelii* strain PKO and observed comparable effects on the emigration of epidermal Langerhans cells. These findings suggest that the immunological mechanisms under

investigation are not limited to a specific *Borrelia* strain and are likely conserved across different *Borrelia* species, including those more commonly transmitted by *Ixodes ricinus* in Europe.

4) In experiments where the cells were treated with both salivary gland extract and *Borrelia* spirochetes, were these two treatments performed simultaneously or was there any pre-incubation step?

Borrelia spirochetes and salivary gland extract mixtures were prepared in advance and directly added to the cells, to mimic natural transmission from tick gut. This information was added to the methods section (page 24).

5) Lines 301-303: In addition to *Amblyomma cajennense* saliva, is there any published evidence regarding the role of *I. ricinus* saliva on cytokines and/or T cell polarization?

Yes, there is published evidence indicating that *I. ricinus* saliva modulates host immune responses by influencing cytokine production and T cell polarization. Specifically, it has been shown to promote anti-inflammatory Th2 responses and the induction of regulatory T cells (Tregs) (Leboulle et al., 2002, DOI: 10.1074/jbc.M111391200 as Ref 41 in the manuscript; review by Blisnick et al., 2019, DOI: 10.3390/vaccines7040148). Further details on specific salivary molecules involved are provided in the discussion section of the manuscript (page 19).

6) Lines 396 and 411: The authors report that the efficiency of moLC differentiation and the purity of isolated fractions were analysed by flow cytometry. What was the resulting efficiency/purity of these cells?

The efficiency of moLC differentiation ranged between 50–80% across all experiments. Samples with differentiation efficiencies below this threshold were excluded from further analysis. We have now included this information in the Methods section for clarity (page 24, Paragraph “Langerhans cell – T cell coculture”).

Formal points:

Line 134 – reference to Fig. 3F is missing in the text.

The reference was added accordingly on page 8.

Line 440 – replace PBMCS with PBMCs.

This mistake was replaced accordingly on page 24.

Reviewer #2

This excellent manuscript from Strobl et al. describes an intriguing mechanism whereby Langerhans cells are phenotypically altered by tick saliva during *Borrelia burgdorferi* infection, which promotes immune tolerance. The key results are supported by robust data generated using multiple complementary methods. These included analysis of human patient samples, validation with an ex vivo human skin model, imaging, flow cytometry, functional assays, in vitro cell coculture, and transcriptomics analysis by scRNA-seq. Overall, the conclusions were highly significant and supported by appropriate analysis of the experimental data.

We thank the reviewer for the positive evaluation of our manuscript and helpful comments.

However, there were a few minor concerns with the manuscript, as outlined below:

1. Figure 1: The experimental Tick Bite experiment seems to be slightly underpowered. In Fig 1D, the % CD1a+ cells were modestly significant, and, more importantly, there appears to be a nonsignificant trend in the CD1a+ cells/mm², which is the most relevant measurement. Increasing the N of this ex vivo experiment will likely result in achieving statistical significance, but a power analysis and additional replicates should be performed to confirm this. It appears the data in Fig 2A-F may be from the same experiment. If so, additional replicates may also improve statistical power for these data.

We agree with the reviewer's evaluation and have repeated the experiments to achieve more biological replicates for experimental tick bites. The new results now reach level of significance for CD1a+ cells/mm². Data was added to Figures 1D and 2 D-F.

2. Figure 2: In Fig. 2G-H, please re-analyze the IF data to provide a geometric mean distance between Langerhans cells (LC) and lymph vessels, rather than using a "close to" cutoff of 20 μ m distance. Additionally, the title of Fig. 2H should be changed to reflect the data, as not all LC are "in" lymph vessels.

We thank the reviewer for the suggestion and for pointing out the erroneous graph title, which we adapted for Fig. 2H. In addition, we have re-analyzed data of representative clinical tick bite samples using Large Language Model (LLM)-based image detection to provide geometric mean distances between LCs (CD207⁺ and lymph vessels (Podoplanin⁺). The new data are included as graph in Suppl. Figure 3C and described in the Results section on page 7.

3. Figures 4 & 6: Please provide the multiplicity of infection (MOI) for the co-culture experiments involving stimulation with *Borrelia burgdorferi* (Bb) and *Staphylococcus aureus* (Sa) in the figure, text, and methods. Also, these experiments are not infections, which involve an in vivo model. Please change this wording to "stimulation" or "co-culture" in both the figure and text.

CFU / optical density information was added in the figure legends and manuscript text (methods section, page 24). In addition, we changed mentions of "infection" to "bacterial stimulation" in Figure 4 and in the manuscript text.

4. Fig 6: The ex vivo lymphoid organoid model is not adequately explained in the manuscript text or in Figure 6. Please include a brief description of the model, which is currently provided in Supplementary Fig 6, in the main body of the text and/or figure to provide sufficient information needed to understand the methodology and interpret the results.

We now provide a more detailed description of the organoid model in the methods section (page 25).

5. Overuse of abbreviations made reading the manuscript somewhat burdensome, particularly the overuse of TB/TS. Please revise the manuscript, figures, and figure legends to spell out "tick bite" and "tick saliva" throughout.

The abbreviations are now spelled out throughout the manuscript text.

6. While use of LC is appropriate in the manuscript body, abbreviations should be avoided in the graphical abstract.

This change to the graphical abstract was made accordingly.

7. Similarly, abbreviations should be avoided, if possible, in figure titles and section/subsection headings.

Changes were made accordingly.

Reviewer #3

The manuscript titled "Human epidermal Langerhans cells induce tolerance and drive immune evasion upon tick-borne pathogen transmission" by Strobl et al. addresses how tick saliva (TS) and tick-borne pathogens may reprogram human Langerhans cells (LCs) to promote immune tolerance. The authors present a range of experimental models, including ex vivo systems and lymphoid tissue organoids, to examine LC migration, chemokine receptor expression, and T cell polarization. They report that TS induces upregulation of IDO1 and IRF4, which are associated with tolerogenic immune responses, while also promoting a shift toward Th2 and regulatory T cell (Treg) polarization. However, while the study provides some insights, several claims are not fully supported by the presented data, critical controls are lacking, and some conclusions require additional

clarification.

We thank the reviewer for the thorough evaluation of our manuscript.

Below are my specific comments for consideration:

Major comments:

1. Title: The title should be rewritten to tone down its claims. The current wording "... and drive immune evasion" implies a level of direct evidence that is not fully supported by the data presented.

We thank the reviewer for this helpful suggestion. We have revised the title to better reflect the scope of our findings and to avoid overstatement. The new title, "Human epidermal Langerhans cells induce tolerance and hamper T cell function upon tick-borne pathogen transmission" more accurately conveys the data presented and avoids implying direct evidence of immune evasion mechanisms.

2. Introduction: On line 54, it is important to note that skin Langerhans cells (LCs) are not derived solely from monocytes. They also originate from embryonic precursors. Please revise this section to provide an accurate description of the origin of LCs.

We thank the reviewer for pointing this out. The section in the manuscript text (page 4) was revised accordingly.

3. Figure 1: Line 77: The authors found a statistically significant reduction in CD1a⁺ cells in the epidermis of the TB group. However, the results for CD207⁺ cells were not statistically different. Given that both CD1a and CD207 are markers for Langerhans cells, the authors should explain this discrepancy.

We thank the reviewer for pointing out this important aspect. In our previous study (Strobl et al., J Clin Invest, 2022, DOI: 10.1172/JCI161188), we reported a reduction in CD207⁺ Langerhans cells following clinical tick bites, without assessing CD1a expression. For the current study, CD1a was selected as the primary Langerhans cell marker due to staining panel constraints. To validate that CD1a and CD207 show comparable patterns, we included CD207 staining in an initial subset of samples (n = 4), which is shown in Supplementary Figure 1B. Based on your comment, we have now expanded this dataset and observe a statistically significant reduction in CD207⁺ cells as well. These additional data have been included in the revised Supplementary Figure 1B and are referenced in the manuscript.

4. Figure 1: Additional staining: Did the authors stain for other antigen-presenting cells (APCs) in the epidermis? It is possible that other APCs, such as inflammatory LCs with lower expression of CD207, may replace steady-state LCs. These cells could have a significant impact on the immune response. The authors are encouraged to stain for HLA, which could provide insights into the presence of other APCs that might compensate for the loss of steady-state LCs.

We thank the reviewer for this valuable suggestion. In line with the hypothesis that other antigen-presenting cells (APCs) may compensate for the loss of steady-state Langerhans cells, we assessed HLA-DR⁺ cells in both our experimental (flow cytometry) and clinical (immunofluorescence on skin sections) cohorts. In both settings, the numbers and percentages of HLA-DR⁺ cells did not show significant changes. These findings are consistent with the possibility that CD1a⁻CD207⁻ APCs may infiltrate the epidermis following Langerhans cell emigration. We agree that further characterization of these potential compensatory APC populations is interesting to further assess in future studies. The corresponding data have been added to Supplementary Figure 1H-I.

5. Figure 1: Clarity issue: The sentence "...This effect was independent of the sampled body site..." is unclear. The authors did not present a comparison of healthy controls (HC) and tick bite (TB) samples at different body sites, making this claim difficult to interpret. Please clarify this statement or provide supporting data.

We thank the reviewer for pointing this out. LCs at different body sites from both healthy and tick bites are depicted in Supplementary Figure 1C. For clarification, we have now removed samples obtained from healthy controls from the graph and show only clinical tick bite samples in a new version of Suppl. Fig. 1C.

6. Figure 1D: In the ex vivo model, the injection of tick saliva (TS) leads to an overall reduction in the frequency of CD1a+ cells. However, when calculated as a percentage per mm², the reduction is not statistically significant. The authors should explain this apparent inconsistency.

We have performed additional experiments with further biological replicates to increase statistical power. The new results now reach level of significance comparing CD1a+ cells/mm² after injection of tick salivary gland extracts vs. control. Data was added to Figures 1D and 2D-F.

7. Figure 2A: Similar to comment 6, here the statistical difference is found only when calculating the percentage per mm², but not in the total percentages of CD1a. The authors should provide an explanation for this observation.

Figure 2A shows a statistically significant increase in dermal CD1a+ cells after clinical tick bites, while the percentage of CD1a+ cells among all DAPI-positive cells was not increased significantly. We hypothesize that other dermal cell types may have increased concomitantly (indicated by increase of DAPI+ cells, and unchanged levels of HLA-DR+ cells (as indicated above)), therefore the proportion of Langerhans cells among all cells may not have increased significantly.

8. Figure 2C: In contrast to the epidermis, dermal Langerhans cells (LCs) show no significant difference in expression of CXCR4 and CCR7 between the healthy control (HC) and tick bite (TB) groups. While this is reasonable for CXCR4, CCR7 expression should still be higher on dermal LCs as they require this receptor to migrate to the lymph node. This is particularly relevant given that CCR7 expression is significantly higher in the TB group in the epidermis. The authors should address this discrepancy.

We thank the reviewer for this insightful comment. We agree that dermal Langerhans cells (LCs) are expected to express CCR7 to facilitate migration to draining lymph nodes. Our initial analysis focused on differences in CCR7 and CXCR4 expression between tick bite (TB) and healthy control (HC) skin. The absence of differential expression in the dermis may reflect the fact that dermal LCs in both HC and TB groups are already in a migratory state.

To further explore this, we compared CCR7 expression between epidermal and dermal LCs within each group (*Reviewer Figure 1*). This analysis revealed a trend toward higher CCR7 expression in dermal LCs, consistent with their migratory phenotype. While these findings support our interpretation, we did

Reviewer Figure 1. Proportion of CXCR4+ and CCR7+ cells among CD1a+ cells in epidermis and dermis of tick bite sites (left panels) and healthy controls (right panels), detected by immunofluorescence staining of skin cryosections. Numbers indicate p-values (paired t-test).

not include the data in the main manuscript due to space constraints, but we are happy to provide it for the reviewers' reference.

9. Figure 2E: In the ex vivo system, CCR7 is not significantly upregulated in either epidermal or dermal Langerhans cells (LCs). This weakens the authors' claim regarding the presence of LCs in the lymphatics. Can the authors provide flow cytometry plots showing CCR7 expression on LCs in the epidermis and dermis?

We have now performed additional experiments with further biological replicates to increase statistical power (N=7). The new results now reach level of significance and were added to Figure 2E.

10. Figure 2G: The staining for Podoplanin and CD207 is not convincing. Higher magnification images are required to demonstrate that Langerhans cells (LCs) are located in the lymphatics. The supplementary data in Figures S3A-B also show no significant differences in CCR7 expression. The authors must provide flow cytometry data, as it is difficult to draw firm conclusions from the immunofluorescence staining alone.

We agree and now provide higher magnification of the images in Suppl. Figure 3D. Regarding flow cytometric analysis of CCR7 expression in Langerhans cells, we don't believe it's technically feasible to dissect LC numbers within lymphatic vessels vs. intra-vascular skin using flow cytometry. We did, however, perform flow cytometry to investigate CCR7-expression on LCs in experimental conditions (primary LCs in experimental TB model, moLCs, CB-LCs) and detected robust up-regulation (Figures 3C and 3F).

11. Figure 3: Although the representative FACS plots in panel B display an upregulation of chemokine receptors in the TS groups compared to HS, there were no significant differences between HS and TS in upregulating CXCR4 and CXCR7 on moLCs. Additionally, no differences were observed in the migration assay for these treatments (Figures 3C–E). The same holds true for the experiments involving BC-LC in Figures 3F–G. This undermines the conclusion that TS causes LC migration, as HS appears to have a similar effect on migratory capacity.

We agree with this limitation and have thus now included additional analyses illustrating differences in HLA-DR, CCR7 and CXR4 expression after stimulation with human vs. tick saliva. We observed significant increase after TS stimulation compared to stimulation with HS. Data was added to Supplementary Figure 4C.

12. Figure 4: The important control group of HS used in the previous experiments is missing in this set of experiments. Please add it.

We have now repeated the experiments with human saliva controls. Data was added to Supplementary Figure 4C.

13. Figure 4C: The authors mention in the text that IRF4 and IDO1 are transcription factors associated with a tolerogenic setting. However, only IDO1 is upregulated, while IRF4 is not. The authors should clarify the implication of this finding. Additionally, the inclusion of the HS control is important to determine whether the upregulation of IDO1 is TS-specific.

We agree with the reviewer regarding the importance of a HS control and have now repeated the experiments with human saliva controls (data was added to Suppl. Fig. 4D). Furthermore, we now show IRF4 and IDO1 expression separately (data was added to Figures 4 C, E and Suppl. Figures 4A, D).

14. Figure 4D: Please also show the levels of IRF4 separately, without combining them with IDO1. Additionally, the co-stimulation with TS and B31 does not significantly increase the expression of IDO1 and IRF4 compared to B31 stimulation alone. Therefore, the statement in lines 182–183, "...an effect that was further increased upon combined stimulation with TS," is incorrect and should be revised.

We now show IRF4 and IDO1 results separately (see above). The incorrect statement was revised, we thank the reviewer for pointing this out.

15. Figure 4F: It appears that TS has no significant effect, as it also fails to reduce the immunogenic response to *Staphylococcus aureus*. The authors should address this point.

We agree and have revised the corresponding manuscript section on page 12 (lines 306 – 308).

16. Lines 191–193: The data presented do not fully justify the conclusions made in these lines. The authors are encouraged to revise this section or provide additional supporting data to strengthen their claims.

We agree and have revised this manuscript section (page 12).

17. Figure 5D: Does the UMAP describing CD207 expression represent the lesional skin sample, the non-lesional skin sample, or an integration of both? The authors should clarify this.

The UMAP shows CD207 expression in all analyzed cells. We have now clarified this in the figure legend.

18. Figure 5E: The number of Langerhans cells (LCs) in the UMAP is quite low, particularly in the lesional skin samples. This raises concerns about the reliability of the analysis presented in this figure. The authors should perform random downsampling to demonstrate that the results are not solely due to the lower LC numbers in the lesional group.

We agree with the reviewer that the number of LCs in Figure 5E is quite low (lesional skin: 116 LCs, non-lesional: 88 LCs). To address this issue, we have included an additional publicly available single-cell RNAseq dataset from erythema migrans patients, in which we see the same polarization of LCs. Data is presented in Supplementary Figure 5B-D.

Minor comments

1. Line 53: Please add the abbreviation "DC" here before using it in line 63.

This change was made accordingly.

2. Fig. 5: Please use the abbreviation "LC" as in the rest of the figures.

This change was made accordingly.

Reviewer #4

In the current study, Strobl J. and Kleissl L., et al., investigated tick saliva (TS)-induced Langerhans cell (LC) activation using human primary samples and multiple experimental models. They conducted multiple immunostaining analyses using human samples to measure LC activation and their migration to the dermal layer and lymph vessels. The hypothesis was further assessed using the ex vivo model; however, there are some questions about the reliability of data using the current model. Additionally, single-cell profiling of primary patient samples was performed to understand the tolerogenic T cell response after TS-induced LC activation. The data are original and novel, and the manuscript is well-written, though some key experimental design details are lacking.

We thank the reviewer for the overall positive evaluation of our manuscript.

The concerns regarding the study are outlined below.

1- Human lymphoid tissue organoid model: While there is a typo in the writing, the authors reported using 150,000 cells to generate organoids. First, this initial cell number seems excessive, raising questions about generating stable aggregation without centrifugation within just 48 hours. Second, such a model cannot be accurately termed an "organoid," since only one cell type is present, and the current model lacks the complexity of recently published immune system organoids. "Spheroid" or "aggregate" would be better terms to describe this model. The authors

referred to the protocol published in Nature Medicine volume 27, pages 125–135 (2021) to generate their model, but key differences between these models should be provided in detail.

We thank the reviewer for pointing this out. The reported cell number is indeed at 1.5×10^6 initially seeded cells. The cited publication used human tonsil, lymph node or spleen tissue to create “organotypic models of lymphoid organs”, as the authors phrase it the original publication from Wagar et al., as well as the follow-up publication from Kastenschmidt et al (doi: 10.1016/j.immuni.2023.06.019). In response to the reviewer’s concerns, we have adapted the phrase “organoid model” to “immune organoid culture model”, to better reflect the low complexity and spatial organization of the model. In addition, we have added further imaging of the model to Supplementary Figure 6B and describe it now in more detail in the methods section (pages 26 – 27).

2- In their model, the authors used “100 U/mL penicillin-streptomycin (Gibco) and 100 µg/mL normocin” in the culture medium, followed by co-incubation with live *B. burgdorferi* (B31) or *S. aureus* bacteria. These antibiotics are not first-line therapies for Lyme disease, but penicillin and normocin have demonstrated effectiveness against *Borrelia*. Therefore, the presence of penicillin-streptomycin and normocin during incubation could compromise the integrity of the data by potentially affecting bacterial viability and host-pathogen interactions.

We thank the reviewer for this important comment. To clarify, antibiotics were *not* present during the initial pulsing of moLCs with either *S. aureus* or *B. burgdorferi* B31. Therefore, antibiotics were not used for any of the experiments where bacteria were present. However, for prolonged co-culture experiments lasting up to one week, Penicillin-Streptomycin (PenStrep) was included in the medium to prevent bacterial overgrowth and associated cytotoxicity. The inclusion of antibiotics in these extended cultures was solely for maintaining cell viability and not intended to assess antimicrobial efficacy or therapeutic response. We have added this clarification to the Methods section on page 25 for transparency.

3- This reviewer acknowledges that investigators have published their ex vivo model previously, but this skin model is not feasible, and the result should be removed as no conclusion could be made based on this model.

We thank the reviewer for the comment on the ex vivo human skin model. We respectfully emphasize that this model, as established in our previous study (Strobl et al., J Clin Invest 2022), has been peer-reviewed and validated as a physiologically relevant tool to investigate early immune responses in human skin. It allows for controlled intradermal application of tick salivary components and pathogens into full-thickness, freshly obtained human skin maintained under organotypic culture conditions. Crucially, the model preserves the native skin architecture and immune cell composition, enabling direct assessment of human-specific responses of resident immune cells – particularly the migration of Langerhans cells (LCs), which was a central focus of this study. Our findings using this system are consistent with LC dynamics observed in clinical biopsy samples, further supporting its validity. While we acknowledge that ex vivo models have inherent limitations, such as the absence of circulation, they provide a valuable and experimentally tractable platform for studying complex host-pathogen interactions in human tissue.

4- The investigators mentioned another model “epidermal sheets” on page 4 and stated that epidermal sheets were cultivated with TS-supplemented media. Where is the detail of this test? What was the concentration? Which cells and from what source were they supplied? Is it primary epidermal cells or a cell line? What media was used? How were the LCs maintained in culture? What was the passage number? Many details are missing here.

We thank the reviewer for pointing this out and apologize for omitting this information. We added a paragraph to the methods section describing the epidermal sheets in detail (page 22).

5- The abstract should be revised, currently it read like a review paper and aims, methodology and results of the study are unclear.

We thank the reviewer for the feedback and revised the abstract to include more information on aims, results and methodology.

6- Figure legends are not appropriate, for example what SHC means in Figure 1?

We renamed “tick saliva” to “salivary gland extracts” (SGE) and revised the figure legends to include all abbreviations used in the figures.

7- Schematics in Figure 1A and 1C are the same for skin. Based on the text, these are meant to represent two different conditions. This should be revised. The same applies to the schematics in Supplementary Figure 1.

Schematics in Figures 1 A+C were included to better differentiate between clinical and experimental tick bite. We adapted the schematics to better represent the different experimental conditions.

8- Suppl.Fig.1 part C the n number should be 8 for each group and not 18, revise accordingly.

We have corrected this mistake.

9- Suppl. Fig. 1F: how did investigators measure the number of LCs in supernatant?

The LC numbers were measured by flow cytometry. We have added this information to the figure legend.

10- The detail of ex vivo human skin TB model should be provided.

We now provide further information on the ex vivo TB model in the methods section of the manuscript (page 21).

11- What was the injection volume for delivering the TS into the skin?

The injection volume for TS was at 30 μ l (concentration of 20.6 μ g/ml). We included this information to the methods section (page 21).

12- Page 4 and Figure 1: In the ex vivo culture of human skin, blood cells are typically removed. Even if the authors did not deplete the blood cells, the number of CD45+ and other blood cells would likely be minimal, raising concerns about the accuracy of cell count. Can the authors clarify how they addressed this limitation?

We thank the reviewer for raising this important point. In our ex vivo model, we used full-thickness human skin, which retains resident immune cells, including LCs. Notably, LCs are not typically present in human peripheral blood, and their presence in our system reflects their tissue residency. To account for potential variability in cell numbers, we normalized our analyses by quantifying cells per square millimeter of tissue, which allows for accurate comparison across conditions regardless of overall cellularity.

13-Page 4 and Figure 1: Another limitation is the limited migration of blood cells (CD45+) from ex vivo skin to the supernatant due to the absence of circulation. Even if those CD45+ cells are migrating from ex vivo skin to the supernatant, the majority are likely originating from the skin edges which are not exposed to the TS. This has potential impact on the results. Can the authors address this limitation?

We thank the reviewer for highlighting this important consideration. To minimize the contribution from skin edges, we used a small size of skin punches and centrally applied tick saliva to ensure even exposure. As pointed out before, LCs are not typically present in the circulation, our primary aim in investigating LCs in supernatant was therefore to exclude undirected emigration from the skin biopsy locally.

14- What was the number of CD207+ cells in the supernatant where no PBS and TS was injected into the skin? What was the number of CD1A+ cells in the supernatant? The investigators used different types of antibodies for same study.

We have now repeated our investigations in supernatant of the skin explant model using both CD207 and CD1a antibodies, and a control without injection of PBS or TS. In line with our initial results, we did not detect differences between groups. Data was included as new Supplementary Figure 1F.

15- Figure 2G: The merged figures should be provided.

We agree and now included an additional representative image with higher magnification and including the merged fluorescence channels as Supplementary Figure 3D.

16- According to the text, the CD207+ cells should be next to the Podoplanin+ cells, but these two antibodies are overlapped, and it seems the exact cells are positive for the same markers. Is there any explanation for this? Otherwise, high-quality, high-magnification images should be provided confirming the proximity of cells without marker overlap. This also may change the conclusions made in part H.

We agree and have included additional images (see response to comment 15, above).

Point-by point response to reviewers

Reviewer #1 (Remarks to the Author):

The authors have addressed all my questions and recommendations raised in the first review. I wish the authors good luck in their future research.

Reviewer #2 (Remarks to the Author):

The reviewer sincerely appreciated the thorough rebuttal of all points raised in response to the initial manuscript review. The new data and revised text have strengthened the manuscript. No additional comments or concerns.

Reviewer #3 (Remarks to the Author):

I have no further comments; the authors have addressed my concerns.

Reviewer #4 (Remarks to the Author):

Authors have addressed some of this reviewer's concerns, however the following concerns remained:

- *In response to the previous concern "such a model cannot be accurately termed an 'organoid,' since only one cell type is present, and the current model lacks the complexity of recently published immune system organoids. 'Spheroid' or 'aggregate' would be better terms to describe this model", authors changed the term "organoid model" to "immune organoid culture model," which is not scientifically correct. Therefore, this term should be revised throughout the manuscript to a more accurate scientific term for their model, such as "aggregate" or "spheroid."*

Author Response: We appreciate the reviewer's suggestion regarding the terminology. While the term "immune organoid" aligns with the nomenclature used by Wagar LE et al. (Nature Medicine 2021), where the protocol for the specific model we are using was first published, we have nevertheless revised the manuscript to use the term "immune spheroid" throughout.

- *Some key experimental details are lacking. For example, they claimed that "We added a paragraph to the methods section describing the epidermal sheets in detail (page 22)." However, there are many details missing here that concern reproducibility of data. I assume they refer to page 21. When did LCs emerge? What do authors specifically mean by "skin reduction surgeries"? What was the anatomical location for those biopsies? The heading refers to isolation of LCs from epidermal sheets—how many days after culture could you harvest enough LCs? Is it 24h? What was the number of LCs that could be detected in the supernatant from 8 mm biopsies after 24h?*

Author Response: Thank you for pointing out these important omissions. We have now revised the methods section (pages 22/23) to clarify the anatomical origin of the skin samples, define "skin reduction surgeries," and include detailed timing (24 hrs). Furthermore, we performed additional experiments on LC emergence and recovery from epidermal sheets (new Supplementary Fig 1H).

- *Page 20: The authors reported that SGE was injected "subepidermally" into skin explants using an insulin syringe. However, it's difficult to claim that the injection was strictly limited to the "subepidermal" region, as there may be leakage into the dermis and potentially the hypodermis.*

Author Response: We agree with the reviewer that strictly limiting injection to the subepidermal space is difficult, and potential leakage may have occurred to lower dermal regions. We therefore addressed this limitation in the methods section of the manuscript (pages 22/23).

- *Page 7 and Figure 2: Authors investigated spatial relations of lymphatic vessel endothelial cells and LCs in the dermis of tick bite-affected skin. They conducted immunostaining for CD207 (Langerhans cell marker) and podoplanin (lymphatic endothelial cell marker), and then measured the percentage of LCs in proximity (distance $\leq 20 \mu\text{m}$) and within lymph vessels. However, the images provided in Figure 2G (supporting the data presented in Fig. 2G) and Supplemental Figure 3 do not differentiate between the two cell types. The image clearly indicates that the same cells were stained for both CD207 and podoplanin, suggesting the antibody was not specific. Therefore, the claim should be further validated.*

Author Response: We thank the reviewer for the important observation regarding the immunostaining in Figure 2G and Supplemental Figure 3. We agree that the images in the initial version of the manuscript did not clearly distinguish between Langerhans cells (CD207⁺) and lymphatic endothelial cells (podoplanin⁺), which raised concerns about antibody specificity. To address this, we repeated the analysis using an alternative staining panel that included CD31 as an additional endothelial marker. This approach allowed us to unambiguously discriminate between endothelial cells and Langerhans cells. Importantly, the results obtained with the CD31-based staining confirmed our original findings regarding the proximity of LCs to vessels. We have updated the manuscript on page 7 and in Figure 3G,H to include these new data and corresponding images, which we believe strengthen the validity of our claim.